# Thoughts and Lessons on Using Visual Foundation Models for Manipulation

**Ryan Chen**                                                           *ryanchen@u.northwestern.edu*
*Department of Statistics and Data Science*
*Northwestern University*

**Ziteng Pang**                                                 *zitengpang2027@u.northwestern.edu*
*Department of Statistics and Data Science*
*Northwestern University*

**Bradly C. Stadie**                                                  *bstadie@northwestern.edu*
*Department of Statistics and Data Science*
*Northwestern University*

**Reviewed on OpenReview:** *https://openreview.net/forum?id=o6mnkDzVuc*

## Abstract

Training vision-based robotic systems from scratch is both computationally expensive and memory intensive. To mitigate these challenges, recent approaches forgo end-to-end training in favor of adopting visual representations from visual foundation models – large scale models designed for broad task transferability. Recent years have seen numerous vision foundation models emerge, including several designed specifically for manipulation tasks. However, we still lack clear principles for what makes these models effective for robotics applications. To address this gap, we systematically evaluate vision foundation models to understand what makes them effective for offline robotic learning. We find that across eleven diverse vision encoders, a representation's ability to reconstruct edges and predict keypoints strongly correlates with its performance on manipulation tasks. Extensive correlation analysis across 21 manipulation tasks consistently shows that representations preserving edge and keypoint information achieve the highest environment success rates. These findings appear to challenge conventional wisdom about holistic reconstruction-based pretraining and offer a new lens for understanding what makes vision representations effective for robotics.

## 1 Introduction

Consider the challenge of a robot learning to manipulate its environment directly from raw images. In contrast to image classification systems, robotic vision systems must understand not only what objects are present, but their physical properties, spatial relationships, and how they will respond to interaction. Training a system from scratch to both understand visual scenes and control actions requires enormous amounts of robot data and computation (Finn et al., 2016). Vision foundation models offer a promising solution to this bottleneck: by leveraging visual representations pretrained on large datasets, robots can start with a strong prior about the visual world. Prior work has shown these pretrained representations can transfer surprisingly well to robotics tasks, even when trained on out-of-domain data (Parisi et al., 2022). Building on this insight, several foundation models have been developed specifically for robotic applications (Radosavovic et al., 2022; Nair et al., 2022; Ma et al., 2022; 2023; Chen et al., 2024).

Despite widespread adoption of pretrained vision representations in robotics, the characteristics that make them effective for control remain largely unclear. While several works have studied different aspects of

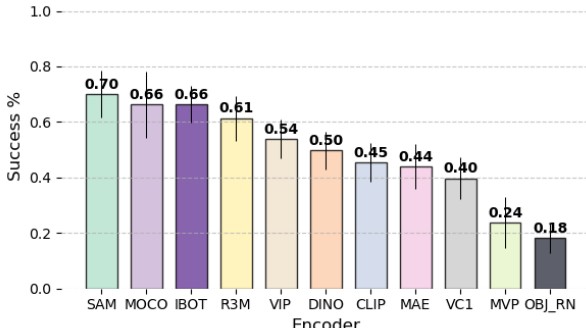

Figure 1: Visual encoders exhibit substantial variability in their effectiveness when deployed in manipulation tasks with imitation learning. We examine downstream auxiliary tasks that can help explain the variability, providing insight on what is important in visual representations when trying to accomplish manipulation tasks.

these representations—from inductive biases (Ze et al., 2023; Yen-Chen et al., 2020) to robustness (Burns et al., 2024) and cross-environment performance (Parisi et al., 2022; Majumdar et al., 2023)—we lack clear principles for selecting and adapting them for manipulation tasks. To address this gap, we conduct a systematic evaluation of nine diverse encoders—ranging from general-purpose to robotics-specific—across 21 manipulation tasks.

Consistent with Majumdar et al. (2023), we find that no single pretrained representation uniformly outperforms others across all tasks. However, while investigating these encoders, we made a curious discovery: there exist several easy-to-compute diagnostic measures of a visual encoder that are strongly correlated with downstream robotic performance. Specifically, a representation's ability to capture accurate keypoints and reconstruct high-fidelity edges shows remarkably strong correlation with manipulation performance (Pearson $\hat{\rho}_r = 0.92$, Spearman $\hat{\rho}_s = 0.88$). Visual analysis using Sobel filters on reconstructions reveals these measures align with a model's ability to precisely locate objects and their boundaries, capabilities that are crucial for manipulation.

Our contributions are as follows:

1. Through extensive evaluation across 21 manipulation tasks, we identify which vision encoders consistently excel at robotic control. Our results show consistency across both behavior cloning and offline reinforcement learning, providing clear guidance for the robotics community on representation selection.

2. We propose easy-to-compute diagnostic measures that strongly correlate with a vision encoder's performance on downstream manipulation tasks. In particular, we show that scene reconstruction and edge reconstruction quality are remarkably associated with manipulation success, even in OOD scenarios.

3. We demonstrate that representations trained with discriminative pretraining objectives consistently outperform those trained with holistic reconstructive objectives. This finding helps resolve an apparent contradiction in the literature between the effectiveness of supervised versus self-supervised pretraining. Namely, we show that the pretraining objective, rather than the source of supervision, is what matters most for downstream robotic control.

## 2 Related Works

**Representations for Robotic Manipulation**   In robotic manipulation, visual pretraining learns transferable features for downstream tasks. Parisi et al. (2022) demonstrated that visual representations can perform as well as ground-truth state representations, showing that using learned representations alone can be highly effective. Thus, with the rise of foundation models (Bommasani et al., 2021), researchers have begun leveraging these large-scale representations to tackle initialization challenges (Finn et al., 2016).

This insight has led to the adoption of general-purpose vision models for learning manipulation skills. Additionally, vision encoders such as MVP, R3M, VIP, and VC1 (Radosavovic et al., 2022; Nair et al., 2022; Ma et al., 2022; Majumdar et al., 2023) capture physical dynamics by training on robotics-adjacent datasets (Shan et al., 2020; Goyal et al., 2017; Grauman et al., 2022). Other approaches incorporate inductive biases for grasping and pose estimation (Ze et al., 2023).

In parallel to these efforts, a growing body of work focuses on learning object-centric visual representations. These approaches, including slot attention mechanism (Locatello et al., 2020; Heravi et al., 2023), and OCLR (Yoon et al., 2023), aim to decompose scenes into structured object representations using self-supervised objectives such as contrastive learning, temporal alignment, and localized masked reconstruction. In addition, entity-centric models like EIT (Haramati et al., 2024) and EC-Diffuser (Qi et al., 2025) also leverage keypoint information on top of learning from pixels, and can produce generalizable visuomotor control policies. Our findings about keypoint correlations are aligned with intuitions behind these object and entity centric designs, however our focus is on frozen holistic pretrained representations.

Meanwhile, some studies (Hansen et al., 2023; Sharma et al., 2023) suggest that training from scratch or fine-tuning pretrained models can also yield strong results. In this work, we evaluate off-the-shelf vision encoders, comparing both manipulation-specific and general-purpose models. Our goal is to develop data-driven metrics to assess representation quality in manipulation tasks, helping guide future advancements in designing pretrained vision models.

**Representation Evaluation and Benchmarking in Robotics**   Benchmarking foundation vision models for robotic tasks has been extensively studied. Hu et al. (2023) identified behavior cloning and inverse reinforcement learning as key paradigms for analyzing visual representations. Several studies (Burns et al., 2024; Schneider et al., 2023; Sax et al., 2021) evaluate out-of-distribution (OOD) performance of vision encoders, while others (Parisi et al., 2022; Hu et al., 2023) focus on in-distribution performance, which aligns closely with our work.

A key discussion in this space concerns self-supervised versus supervised vision pretraining and their effectiveness for manipulation tasks (Parisi et al., 2022; Burns et al., 2024). Majumdar et al. (2023) found that no single representation consistently outperforms others and introduced the VC-1 encoder, which we include in our evaluation.

Beyond benchmarking, researchers have explored the characteristics of effective representations. Wulfmeier et al. (2021) studied how dimensionality and disentanglement impact multitask performance, while Tomar et al. (2023) analyzed the role of auxiliary training objectives. Both studies, along with Majumdar et al. (2023), suggest that no universal learning algorithm or representation excels across all robotic tasks.

Another line of work highlights the benefits of inductive biases. Burns et al. (2024); Yen-Chen et al. (2020) demonstrated that segmentation-oriented inductive biases improve performance in robotics tasks. To this end, we include SAM (Kirillov et al., 2023) in our evaluation suite. Our work builds on these efforts by identifying key representation qualities that influence manipulation performance.

## 3 Background

To evaluate vision representations for robotics, we need methods that can effectively learn from offline datasets of expert demonstrations. We focus on two such approaches: behavior cloning, which directly

mimics expert actions, and offline reinforcement learning, which learns a value function from demonstrations. In both cases, we maintain a simple architecture (Figure 2) where states and goals are processed through the vision encoder being evaluated.

**Behavior Cloning**  Behavior cloning seeks to approximate expert behavior by learning a state-conditional action distribution $\pi_\theta^{BC}(a|s)$. We implement this by training a three-layer MLP to regress expert actions onto scene embeddings produced by the vision encoder $\phi$. Specifically, for state-action pairs $(s_E, a_E)$ sampled from the expert dataset $D_E$, we minimize $\min_\theta \mathbb{E}_{a_E, s_E \sim D_E} \|\pi_\theta^{BC}(a|\phi(s_E)) - a_E\|_2^2$. The state space consists of image embeddings augmented with goal image embeddings, excluding proprioception to focus our evaluation on visual representations.

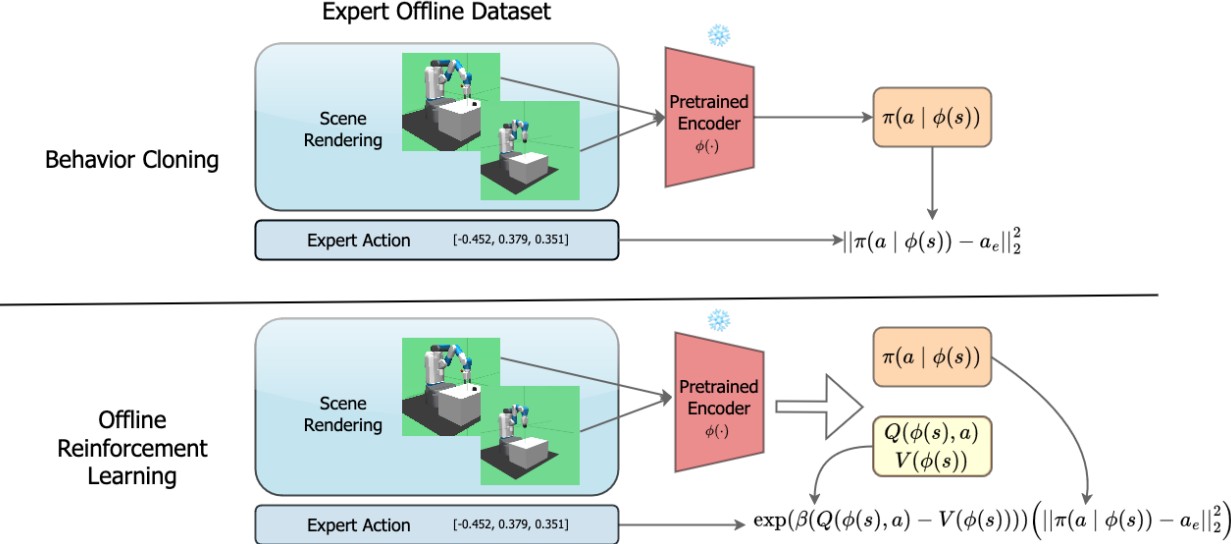

Figure 2: **Imitation algorithms.** We evaluate visual representations using offline imitation methods that learn from expert demonstrations. After training on expert datasets, policies $\pi(a|\phi(s))$ are evaluated in their respective environments.

**Offline Reinforcement Learning**  We also evaluate representations using implicit $Q$-Learning (IQL) (Kostrikov et al., 2022). To maintain consistency with our behavior cloning experiments and focus purely on evaluating visual representations, we use inverse $Q$-Learning (Garg et al., 2021) to estimate $Q$-values directly from expert demonstrations without reward signals, after which implicit $Q$-learning estimates its $V$ function. This creates a reward-agnostic implicit $Q$-learning algorithm. Like our behavior cloning experiments, we exclude proprioception data and train only on visual representations. While other offline reinforcement learning methods exist, most require either reward signals (e.g., CQL (Kumar et al., 2020), SAC-n (An et al., 2021)), or environment interaction (e.g., GAIL (Ho & Ermon, 2016)), making them unsuitable for our evaluation framework.

## 3.1 Environments

We aim to study the effectiveness of pretrained visual representations in how they encapsulate information about any given scene, such as the acting agent, the objects in the environment, and the goal of the tasks. To this end, we select environments based on the following criteria: **(a)** they must be renderable into images to be encoded by visual representations; **(b)** solvable using only visual representations without proprioception data ensuring that the agent acts solely on visual representations; **(c)** encompass a diverse range of tasks with varying complexities, such as grasping, nudging, or hand gripping; and **(d)** have variable goal states and starting configurations to promote generality in the analysis.

With these criteria in mind, we arrive at 21 robotic manipulation tasks from the Fetch Suite (Plappert et al., 2018), AdroitHand Suite Rajeswaran et al. (2017), and Metaworld Suite (Yu et al., 2020). We created experts that could solve these three environments with at least 95% success rate. These experts provided offline datasets for our behavior cloning and offline reinforcement learning experiments. Details about the environments and experts are discussed in Appendix A. Examples of environment renderings are shown in Figure 3.

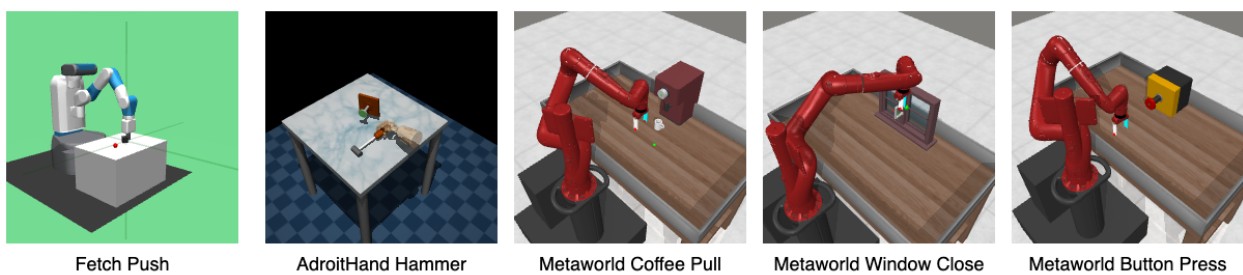

Figure 3: A sample rendering of some of the environments in our evaluation suite.

### 3.2 Vision Encoders

We evaluate nine off-the-shelf, frozen vision encoders, each encoding $224 \times 224$ RGB renderings of environment scenes into vector representations. We include both models specifically trained for manipulation tasks and foundation models trained on internet-scale datasets. The selected encoders span diverse pretraining objectives, from prediction and distillation to contrastive and holistic reconstruction approaches. A full list of considered encoders and their pretraining objectives and data sets can be found in Table 6.

A fundamental aspect of vision encoders is the structure of the representations they generate. Many models produce single-vector representations for an entire image. In contrast, models such as SAM generate spatial feature maps corresponding to VIT patches, and are pooled with convolutions to produce a holistic scene representation. These structured representations often benefit from transformer architectures for downstream tasks, for which we consider in Appendix I. The remaining encoders in our evaluation suite produced a single holistic vector representation for each 224x224 RGB image.

Given the spatial structure of SAM's embeddings, we initially applied a transformer-based policy to learn actions. However, as shown in Appendix I, the MLP policy and attention policy performance did not significantly differ from each other. Since the transformer policy did not provide a significant advantage, we opted to use the MLP policy for consistency across all encoders, ensuring a fair evaluation framework. The use of an MLP policy is well-established in the literature as an effective method for assessing embedding quality (Kumar et al., 2022). By maintaining a standardized evaluation methodology, our results facilitate a direct comparison of the relative effectiveness of different vision encoders for manipulation tasks.

Another source of variation among the encoders is the dataset used for pretraining. While all encoders were pretrained on different real-world data sets, the evaluation was conducted in simulation. Parisi et al. (2022) demonstrated that differences in pretraining data do not influence downstream simulation performance. This suggests that the encoder performance can be directly compared despite difference in training data.

## 4 Experimental Investigation of Vision Encoders

In this section, we evaluate the performance of different vision encoders across a diverse set of robotic tasks. We focus on understanding how different pretraining strategies, task complexities, and offline learning paradigms impact the success rates of policies for robotic tasks.

### 4.1 Behavioral Cloning

We evaluated behavior cloning across 21 environments by regressing expert actions onto image embeddings from the vision encoders. For each policy, we pass both the current scene and goal state images through the encoder. The resulting embeddings are concatenated and processed through a three-layer MLP with hidden layers [256, 128, 64] to predict a four-dimensional action vector. We trained policies using 2000 expert trajectories per environment. Additional training details can be found in Appendix B.

In Figure 1, we see that SAM representations do the best on average, even surpassing the average success rates on policies pretrained with manipulation-based representations. In agreement with Hu et al. (2023), we note that the ResNet50-based representations, (R3M, VIP, MOCOv3) are quite competitive, surpassing the success rates of all ViT representations except for those of SAM representations.

**Representations for Manipulation are more Sample Efficient.** Data efficiency has always been an area of interest for robotics and in general imitation learning. Particularly for robotics, large-scale high quality data collection is time-consuming, and logistically complex. Thus finding a representation that does well in a small sample regime is useful.

To study sample efficiency, we choose environments where the average success rate was at least 80 percent across all representations in our behavior cloning evaluation. We identified five environments: button press, window open, plate slide, door close, and window close. For almost all representations, these environments can be solved with near perfect accuracy with as little as 40 demonstrations. We examine performance after training using 5, 10 and 40 expert demonstrations.

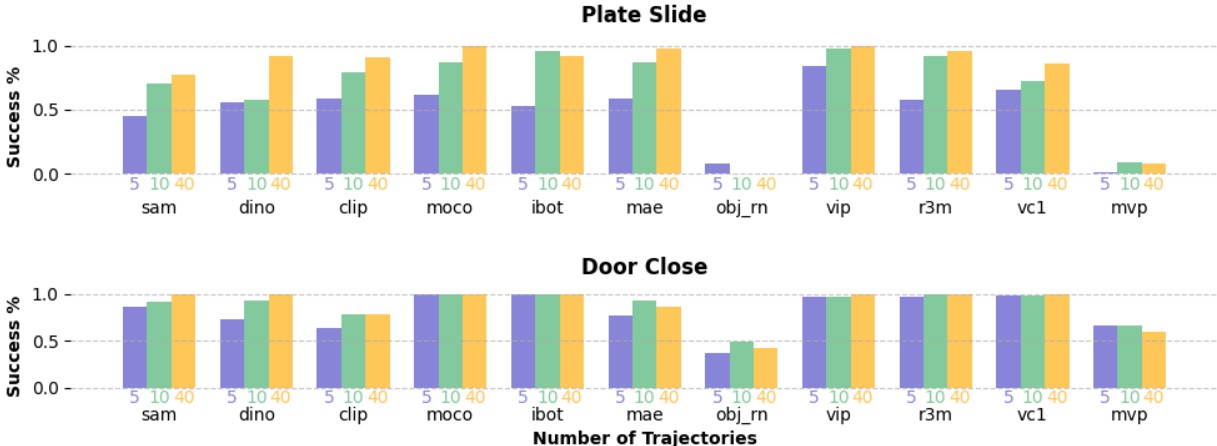

Figure 4: **Sample efficiency of representations.** Behavior cloning trained on 5, 10, and 40 expert demonstrations. Representations trained for manipulation such as R3M, VIP are more sample efficient in general as they achieve higher success rates even with just 5 expert demonstrations. MoCov3 representations are also competitive in the low sample regime. Sample efficiency plots for button-press, window-open, and window-close are shown in Appendix D

From Figure 4, we can see that generally, pretrained representations for manipulation are more sample efficient. MoCov3 is the most sample efficient relative to general pretrained representations, as it eventually solves the environments with 100 percent success using 40 trajectories. However it is not as sample efficient as R3M, or VIP since these representations can solve the environment at a higher success rate with fewer expert trajectories. In general, representations trained specifically for manipulation tasks are more effective when sample sizes are small, which is aligned with manipulation pretraining literature (Radosavovic et al., 2022; Nair et al., 2022).

**OOD performance correlates with ID performance**  In robotics, it is common for the visual scene to change, due to distractions entering the scene, or a change in lighting intensity during a trajectory. In the context of visuo-motor control, it is important to identify image embeddings that are robust such visual scene perturbations. To this end, we study the out of distribution performance of the vision encoders in the Gymansium environments by considering the following out of distribution scenarios: 1) a nuisance object in the background at a random location, 2) random lighting intensities, 3) and a combination of both nuisance object and lighting intensity changes. These perturbations are also studied in Burns et al. (2024). Figure 5 compares the out of distribution performance of the different encoders across the listed OOD scenarios with their ID performance. The correlation between ID behavior cloning and OOD behavior cloning has a Spearman correlation of $\hat{r}_S = 0.642$ and a Pearson correlation of $\hat{\rho}_r = 0.673$. The performance distributions with each type of OOD setting is presented in Appendix H.

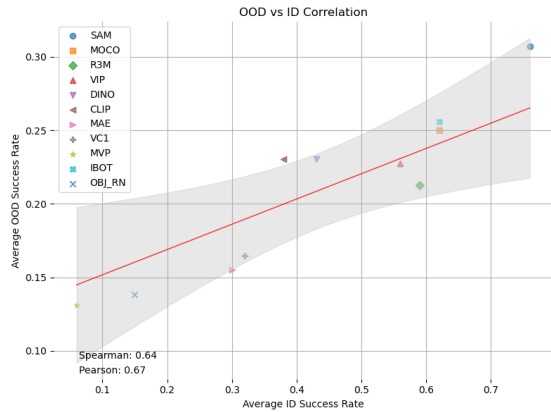

Figure 5: **OOD success rates are overall aligned with ID scenarios**. Stronger performing encoders in the ID case also perform relatively stronger in the OOD scenario.

**Performance distribution is architecture agnostic**  Both ResNet50 and ViT architectures were evaluated across all tasks. As shown in Figure 1, high-performing models exist in both architectures: ResNet-based models like VIP, R3M, and MOCO perform well, as do ViT-based models like SAM and IBOT. Similarly, lower-performing models appear in both architectures, with ResNet-based OBJ RN and ViT-based MVP and MAE showing relatively weaker performance. This distribution suggests no clear performance advantage between ViT and ResNet-based architectures.

| Architecture | Success % | 1SD |
|---:|---:|---:|
| VIT | 0.484 | 0.145 |
| ResNet50 | 0.498 | 0.188 |

Table 1: Comparative analysis of architecture types show that architecture type does not affect manipulation performance.

**Performance distribution is dependent on pretraining objective**  Figure 1 suggests an association between encoder performance in manipulation tasks and their respective pretraining objectives. Encoders pretrained with holistic reconstruction objectives, specifically VC1, MAE, MVP, and OBJ RN, utilize mean square error loss on pixels, which are aggregated across entire image scenes. These encoders also demonstrate a notably worse performance in manipulation tasks. As discussed in Section 3.2, despite the variety of pretraining datasets, Parisi et al. (2022) has demonstrated that pretraining data set is not critical to downstream manipulation performance. Thus, the comparative analysis in Table 2 can be viewed as a counterfactual comparison of pretraining objectives. We further discuss the association between performance and pretraining in Section 5.3. Table 2 also indicates that for models trained with a holistic reconstruction objective, VIT models have a slight edge over ResNet based models for manipulation tasks, however the performance gains see larger increase when switching from a holistic reconstruction objective, to a discriminative pretraining objective.

| Architecture | Objective | Success % | Standard Deviation |
|---|---|---|---|
| ResNet | Holistic Reconstruction | 0.180 | 0.021 |
| ResNet | Discriminative | 0.603 | 0.049 |
| VIT | Holistic Reconstruction | 0.360 | 0.086 |
| VIT | Discriminative | 0.578 | 0.105 |

Table 2: Comparative analysis of pretraining objectives show that discriminative pretraining objectives produce representations that are useful for manipulation.

## 4.2 Behavior Cloning versus Offline Reinforcement Learning

In addition to behavioral cloning, we also evaluate the effectiveness of various visual encoders when using offline reinforcement learning. Specifically, we use our reward-agnostic version of implicit $Q$-learning, training each policy for 10,000 gradient steps on 2,000 expert trajectories. We chose these parameters to make success rates comparable with our behavior cloning results. Training details and the algorithm description can be found in Appendix B and Section 3 respectively.

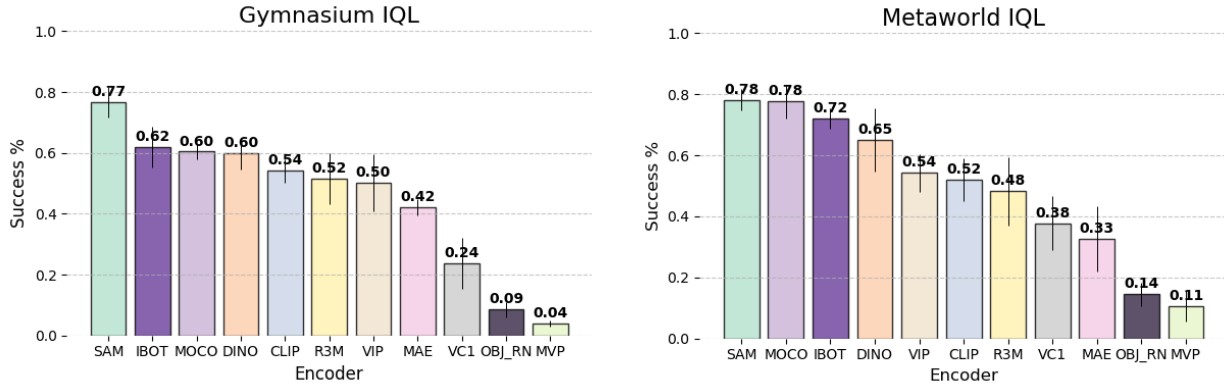

Figure 6: **Success rates with offline reinforcement learning with different representations.** In agreement with Figure 1, we see that the representations with reconstruction pretraining objectives perform worse.

A surprising pattern emerges when we compare behavior cloning and offline reinforcement learning performance across different types of pretrained representations. While one might expect offline reinforcement learning to consistently improve upon behavior cloning, we find this benefit depends strongly on how the representation was pretrained.

**Holistic reconstruction-based representations struggle with offline RL**   Models pretrained with a holistic reconstruction objective show inconsistent benefits from offline reinforcement learning–for example MAE improves by 12% on Gymnasium tasks but declines by 11% on Metaworld tasks, while MVP, and VC1 both see lower success rates by using offline reinforcement learning. In contrast, representations trained with contrastive objectives (DINO, MOCOv3, CLIP) show consistent improvements under offline reinforcement learning.

**Manipulation-specific representations prefer behavior cloning**   Even more surprisingly, representations specifically pretrained for manipulation (VC1, VIP, MVP, R3M) perform worse with offline reinforcement learning compared to behavior cloning as shown in Figure 7. This effect is particularly pronounced for R3M, which excels at behavior cloning but struggles with offline reinforcement learning, consistent with findings from Hu et al. (2023).

There has been much recent investigation on when one should use behavior cloning and when one should use offline reinforcement learning, and the relative strengths of both methods with respect to the expert occupancy distribution being mode covering or mode seeking (Ke et al., 2021; Ghasemipour et al., 2020). In Appendix F, we carry out a small investigation of settings where offline reinforcement learning and behavior cloning excel and struggle. For the purpose of our main findings, we see that there is no large difference in encoder performances due to occupancy mode behavior, regardless of which training method we select.

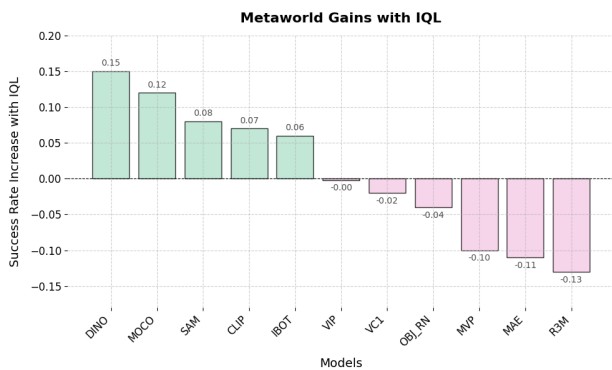

Figure 7: **Holistic representations trained for manipulation perform better using behavior cloning.** In fact, aside from VIP which sees no gain, the manipulation representations achieve lower rates of success. Difference plots for Gymnasium and Metaworld are shown side by side in Appendix C

# 5 What Qualities of Visual Representations Matter for Manipulation?

The goal of this section is to investigate the metrics that correlate[1] with how well visual representations can perform in manipulation tasks. We propose two types metrics: performance in keypoint prediction and performance in image reconstruction.

**Keypoint prediction** is a discriminative task where the representations are probed for information on keypoints such as positions of arm joints, end effectors, objects, or goal locations in the scene. We define keypoint prediction as modeling $P(\mathbf{v}|\phi(S))$, where $\phi(S)$ represents the encoded scene $S$ from the expert demonstration and $\mathbf{v}$ represents the vector of keypoints, which can include goals positions, arm and effector positions and velocities, and scene object positions. Representations excelling in keypoint prediction highlight that the features embed task-relevant information like spatial relationships among key objects and understanding precise manipulation dynamics. This aligns with prior work that leverages keypoints directly for visuomotor control, either by learning keypoint-based representations end-to-end (Boney et al., 2022; Daniel & Tamar, 2022), or by extracting keypoints through pretrained language-guided models to support downstream manipulation tasks (Fang et al., 2024; Palo & Johns, 2024).

**Holistic reconstruction tasks** focus on using global scene representations to reconstruct current scenes or scenes $k$-steps into the future. We define reconstruction tasks to model $P(S|\phi(S))$. That is, we learn to recover the input distribution of scenes that are encoded by the encoder $\phi(S)$. Representations that excel at high-fidelity scene reconstruction suggest that they encode information around scene understanding and model transitions. We pay particular attention to edge reconstruction. Our hypothesis is that representations capable of accurately reconstructing fine grained edges, especially that of objects involved in manipulation, are likely to encode spatial structure that is useful for control.

To evaluate our hypothesis that both metrics are correlated with imitation learning success, we designed a suite of experiments to assess these predictive and reconstructive capabilities of various vision encoders. These experiments test the ability of encoders to extract task-relevant features and reconstruct high-fidelity image representations of current and future scenes.

## 5.1 Keypoint Prediction is Correlated with Success Rates

Following Hu et al. (2023), we approach keypoint prediction with the linear probing method. We took the representations of 100 Metaworld trajectories, each rendered with 150 scenes. For each scene, the linear

---

[1]We use correlation as a measure of how useful a particular metric is, in providing diagnostics of visual representations for manipulation tasks.

probes learned the positions and rotations of the arm joints, and the positions of objects within the scenes, such as hammers, blocks, walls, and windows. The probe also learned the goal positions for each trajectory. Each linear probe uses a linear head and minimizes square error of the keypoint position vectors.

Figure 8 shows an overall negative correlation between the log of validation loss and the behavior cloning success rates for each vision representation. The ability to predict the goal position shows a strong negative correlation between its log of prediction loss and behavior cloning success rate ($\hat{\rho}_s = -0.927$ for Spearman, $\hat{\rho}_r = -0.979$ for Pearson). Meanwhile, predicting the arm's key positions showed a slightly weaker correlation ($\hat{\rho}_s = -0.891$ for Spearman, $\hat{\rho}_r = -0.938$ for Pearson). However, while slightly smaller in magnitude, these relationships were still strongly negative. That the arm positions had the lowest correlation is expected, as successful task completion can be achieved through multiple valid arm configurations—the same task goal can be reached with different joint angles and positions. In contrast, object and goal positions in the scene have a more direct, one-to-one relationship with task success, as they uniquely define the desired end state of a task. When correlating with the OOD success rates, the correlations remain strong, ($\hat{\rho}_s = -0.750$ for Spearman, $\hat{\rho}_r = -0.790$ for Pearson). The graphs showing these correlations are presented in Appendix H.

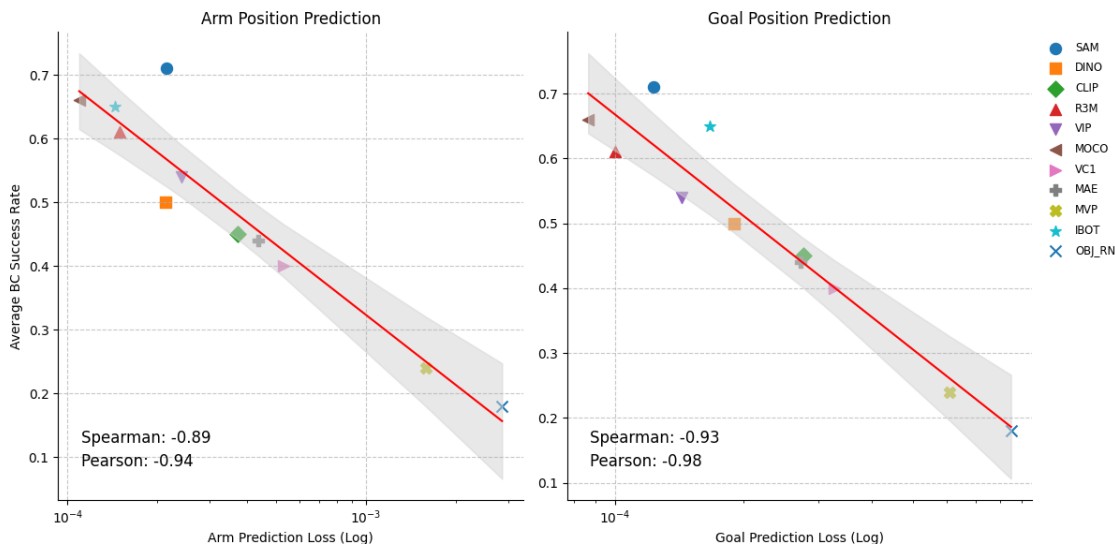

Figure 8: **Correlation between key point prediction validation loss and behavior cloning performance.** Lower validation loss in predicting key points correlates with higher behavior cloning success rates. Reconstruction-based encoders (MAE, VC1, and MVP) show both higher prediction losses and lower success rates. Additional correlations for object position prediction and combined key point positions, along with outlier analysis, are provided in Appendix E.

These correlations suggest that representations that are better at predicting positions in a scene tend to achieve higher success rates in behavior cloning. To verify the robustness of this relationship, we conducted additional analyses to measure the sensitivity of these correlative values. Even when excluding SAM from our analysis, the correlations remain strong ($\hat{\rho}_s = -0.857$ and $\hat{\rho}_r = -0.903$; see Appendix E). Furthermore, a DFBETA analysis (Kleinbaum et al., 1988) confirms that SAM does not unduly influence these correlations. These results strongly support the finding that a representation's position prediction ability, measured by log of validation loss, correlates with its behavior cloning performance.

## 5.2 Reconstructive Abilities are Correlated with Environment Success Rates

To evaluate the reconstructive abilities of various encoders, we encode $224 \times 224$ RGB images from expert trajectories and attempt to reconstruct the current scene as well as scenes 5 and 20 steps into the future. The reconstruction network uses four transposed convolution layers and combines square error loss with

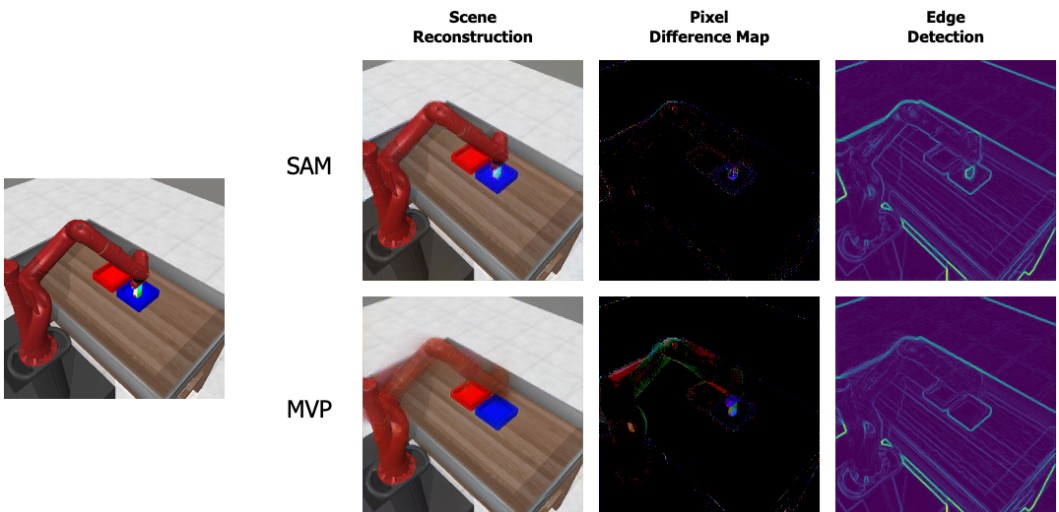

Figure 9: **Scene reconstruction quality correlates with representation performance**. Visualizations show reconstructed scenes, pixel-wise error maps, and Sobel edge detection for SAM (top) and MVP (bottom). SAM, which was the best-performing model, produces accurate reconstructions with clear edges. In contrast, MVP fails to capture critical scene elements.

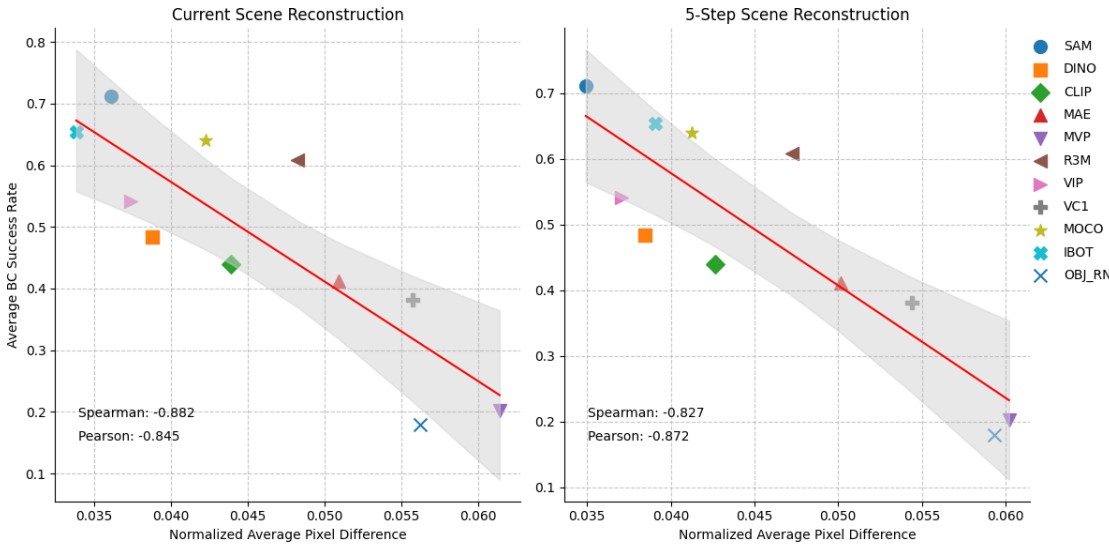

Figure 10: **Scene reconstruction quality correlated with behavior cloning success rate.** Global scene reconstruction is strongly correlated with behavior cloning success rates. Each representation achieves a similar level of reconstruction quality across the current and five-step scene reconstructions. This trend generalizes to the longer horizon, 20-step future scene reconstruction, which is shown in Appendix G.

VGG-based perceptual loss Johnson et al. (2016). The perceptual loss proved crucial. No representation achieved high-fidelity reconstructions without it.

We evaluate reconstruction quality using two metrics: pixel-wise distance and edge preservation via Sobel edge detection (Kanopoulos et al., 1988), between reconstructions and original scenes. Figure 10 shows that behavior cloning success rates correlate strongly and negatively with pixel differences when reconstructing the current scene (Spearman correlation $\hat{\rho}_s = -0.882$ and Pearson correlation $\hat{\rho}_r = -0.845$). In the OOD setting, the correlations are also strong (Spearman correlation $\hat{\rho}_s = -0.918$ and Pearson correlation $\hat{\rho}_r = -0.910$).

The graphs showing these correlations are presented in Appendix H. Similar strong correlations also hold for 5-step and 20-step future scene reconstructions, with analyses available in Appendix G.

The relationship between reconstruction quality and model performance is visually apparent when comparing our best and worst-performing models, demonstrated in Figure 9. SAM's reconstruction shows sharp edges around the block and end effector, with low pixel-wise errors. In contrast, MVP's reconstruction fails to capture these critical scene elements, showing high pixel-wise errors and poorly defined edges. Surprisingly, despite their reconstruction-based pretraining, MAE, MVP, and VC1 representations perform worse at reconstruction than other pretrained models, which we discuss further in Section 5.3. Additional reconstruction examples are shown in Appendix G.

Moving to our next experiment, we see reconstruction of edges is also correlated with success rates. To evaluate edge reconstruction quality, following Kirillov et al. (2023) we apply Sobel filters to both original and reconstructed current scenes, comparing their MSE and structural similarity index measure (SSIM) (Wang et al., 2004). We observe strong negative correlations between edge reconstruction loss and behavior cloning success rates in Figure 11 (Pearson $\hat{\rho}_r = -0.890$ and Spearman $\hat{\rho}_s = -0.918$). The edge structural similarity shows similarly strong correlations (Pearson $\hat{\rho}_r = -0.913$, Spearman $\hat{\rho}_s = -0.927$). This suggests that the ability to encode edge information is particularly important for downstream manipulation tasks. In the OOD setting, the correlations with edge reconstruction is strong (Pearson $\hat{\rho}_r = -0.962$, Spearman $\hat{\rho}_s = -0.910$). Additionally, the OOD correlation with edge structural similarity is also strong (Pearson $\hat{\rho}_r = 0.934$, Spearman $\hat{\rho}_s = 0.955$). The graphs showing these correlations are presented in Appendix H.

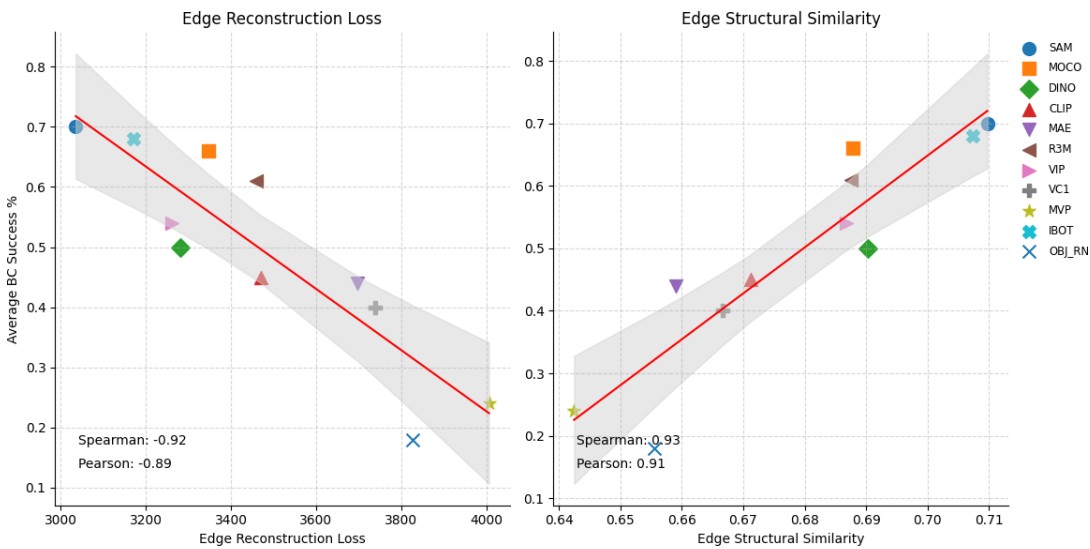

Figure 11: **Edge reconstruction quality correlates with manipulation performance**. Both edge reconstruction error (left) and structural similarity (right) show strong correlations with behavior cloning success rates. Interestingly, reconstruction-pretrained models (MAE, MVP, VC1) consistently show poor edge preservation.

Our analysis of both keypoint prediction and reconstruction tasks reveals a consistent pattern: representations that can predict keypoints with lower validation loss and reconstruct scenes with lower errors achieve better manipulation performance. Notably, reconstruction-pretrained representations struggle with both keypoint prediction and scene reconstruction fidelity. Meanwhile, representations trained with discriminative objectives excel at both tasks. These results suggest that effective robotics representations should perform well on both discriminative and generative auxiliary tasks.

### 5.3 Understanding the Impact of Pretraining Objectives on Robotic Performance

There is an apparent contradiction in the vision-based robotics literature. Parisi et al. (2022) showed that the self-supervised MoCo representations outperforms supervised ones like CLIP and ImageNet pretrained ResNets, in robotic manipulation. Yet, Burns et al. (2024) found supervised representations on ImageNet performed strongest. We propose that this apparent conflict can be resolved by reframing the comparison: rather than supervised versus self-supervised, the key distinction lies in whether representations are pretrained with reconstructive or discriminative objectives.

To understand this distinction, we need to examine how these pretraining objectives fundamentally differ. Holistic reconstruction approaches, like masked autoencoding on the entire image, focus on rebuilding the global view of the image pixel by pixel, often using masked patches. Although masking creates regularized global representations, the resulting representations are embedded with inherent global reconstruction errors. These errors force downstream manipulation policies to learn from reconstruction artifacts rather than focusing on object or region-specific information. In contrast, discriminative approaches optimize representation spaces to capture semantic relationships, either through direct prediction of supervised signals or through contrastive learning between positive and negative pairs (Geiping et al., 2023), facilitating downstream policy learning. Table 3 classifies common vision encoders along these lines.

| Encoder | Objective | Type |
| --- | --- | --- |
| SAM | Segmentation Prediction | Supervised |
| MOCOv3 | Self Contrastive | Self-Supervised |
| R3M | Time Language Contrastive | Self-Supervised |
| VIP | Value function prediction | Supervised |
| DINOv2 | Self-distillation | Self-Supervised |
| CLIP | Text Image Contrastive | Supervised |
| IBOT | Masked contrastive + self distillation | Self-Supervised |
| MAE | **Masked Image Reconstruction** | Self-Supervised |
| VC1 | **Masked Image Reconstruction** | Self-Supervised |
| MVP | **Masked Image Reconstruction** | Self-Supervised |
| OBJ RN | **Image Reconstruction** | Self-Supervised |

Table 3: Pretraining objectives of vision encoders. We provide additional details such as the pretraining data set and architectures used in Appendix B.1

When we analyze our results through this lens of holistic reconstruction versus discriminative pretraining, a clear pattern emerges. Across our experiments, representations trained with discriminative objectives, whether supervised or self-supervised, consistently outperform those trained with global reconstructive objectives. Most surprisingly, reconstruction-pretrained models (MAE, MVP, VC1) perform poorly even at reconstruction tasks. Despite being explicitly trained to reconstruct entire images, these models struggle to capture fine-grained details and edge information critical for manipulation as seen in Figures 10 and 11

We hypothesize this performance gap likely stems from fundamental differences in how these objectives structure representation spaces. Discriminative approaches naturally emphasize meaningful feature separation and semantic understanding. In contrast, global reconstructive approaches focus on overall pixel-level fidelity across the entire scene, potentially at the cost of semantic abstraction of individual objects. This distinction helps explain the strong performance of models like SAM and DINOv2, whose pretraining regimes are known to produce features that facilitate segmentation (Kirillov et al., 2023; Oquab et al., 2023). Similarly, data augmentations and contrastive learning in MoCOv3 and DINOv2 help encode localized features (Chen et al., 2021; Oquab et al., 2023), which proves valuable for manipulation tasks.

## 6 Conclusion

In this work, we conducted a systematic evaluation of pretrained vision representations for robotic manipulation, examining their performance across 21 tasks in three environments. Our analysis reveals that the distinction between global reconstruction and discriminative pretraining objectives is more crucial than the supervised versus self-supervised categorization. Representations trained with discriminative objectives consistently outperform those trained with global reconstructive losses across both behavior cloning and offline reinforcement learning tasks.

We identified three diagnostic metrics—keypoint prediction, scene reconstruction, and edge preservation—that strongly correlate with a representation's manipulation performance. In particular, scene reconstruction and edge preservation are both effective diagnostics for manipulation success, even in the presence of out of distribution perturbations to the environment. Surprisingly, models pretrained with reconstructive objectives struggled with reconstruction tasks, suggesting these pretraining objectives may not effectively capture the fine-grained details needed for manipulation. These findings help explain seemingly contradictory results in prior work comparing supervised and self-supervised representations.

Based on these findings, we hypothesize that future vision encoders for robotics might benefit from combining discriminative objectives with carefully designed reconstruction losses that preserve structural information. For example, incorporating perceptual losses or explicit edge preservation during pretraining might help capture the fine-grained details that prove crucial for manipulation tasks. Additionally, object-centric representations may offer advantages for manipulation tasks, as they capture semantic abstractions of objects within scenes while simultaneously maintaining high-level fidelity for image reconstruction. The landscape of vision encoders is vast, and these insights offer promising directions for further exploration.

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

## A   Environments and Experts

**Metaworld**   The Metaworld suite contains multiple grasping and nudging tasks, in which an end-effector manipulates objects in the environment to complete the task. Metaworld provides expert policies, from which we created expert trajectories for imitation learning. We used a total of 17 environments from Metaworld. Each task intializes to a random starting position, and a random goal position, making this a good environment for multi-goal tasks.

**Fetch Suite**   From the Fetch suite, we evaluate two environments, FetchPush and FetchPickAndPlace. In both environments, the goal is to move a block to a goal, either by pushing it around the table top, or by grasping the block and moving to a goal location. We trained a SAC policy with hindsight relabeling to create the experts.

**AdroitHand**   From the AdroitHand suite, we evaluated two environments, Hammer and Door. Both environments involve manipulating a 30+ DoF hand to open a door or drive a nail into a board with a hammer. We trained a behavior cloning policy on the Farama expert trajectories to create a larger expert trajectory data set. The Fetch and AdroitHand suite are collectively referred to as the Gymnasium environments due to their use of the Farama gymnasium library.

## B  Training Parameters

Each epoch in behavior cloning makes 100 gradient steps using 256 scenes for each gradient step, for a total of 10,000 gradient steps. IQL trains with 10,000 gradient steps using 256 scenes for each step. Our IQL algorithm uses 256 samples of the transition tuple $(s_t, a, s_{t+1})$ (without reward signals), while behavior cloning uses 256 samples in minibatches of $(s_t, a_t)$.

|  | Behavior Cloning | IQL |
|---|---|---|
| Batch Size | 256 | 256 |
| Training Size | 2000 | 2000 |
| Epochs | 100 | - |
| Minibatches | 100 | - |
| Gradient Steps | - | 10000 |
| Actor LR | 0.0008 | 0.00015 |
| $Q$ LR | - | 0.0003 |
| $V$ LR | - | 0.0003 |
| Expectile $\tau$ | - | 0.7 |
| $\beta$ | - | 3 |
| Polyak $\tau$ | - | 0.05 |

Table 4: IQL and Behavior Cloning make the same number of gradient steps on the same size of minibatch.

We do not include any reward signal to perform the offline reinforcement learning. Instead, we use an inverse Q learning method from Garg et al. (2021).

### B.1  Model Architectures

| Encoder | Objective | Type | Pretraining Data | Architecture |
|---|---|---|---|---|
| SAM | Segmentation Mask Prediction | Supervised | SA-1.1B | ViT-B |
| DINOv2 | Self-distillation | Self-Supervised | ImageNet | ViT-B |
| CLIP | Text Image Contrastive | Supervised | WIT | ViT-B |
| MAE | Masked Image Reconstruction | Self-Supervised | ImageNet | ViT-B |
| VIP | Value function prediction | Supervised | Ego4D | ResNet50 |
| VC1 | Masked Image Reconstruction | Self-Supervised | Ego Suite | ViT-B |
| R3M | Time Language Contrastive | Supervised | Ego4D | ResNet50 |
| MVP | Masked Image Reconstruction | Self-Supervised | ImageNet, Robotics Adjacent | ViT-B |
| MOCOv3 | Contrastive | Self-Supervised | ImageNet | ResNet50 |
| IBOT | Contrastive + Self-Distillation | Self-Supervised | ImageNet | ViT-B |
| OBJ RN | Reconstruction | Self-Supervised | Objects 365 | ResNet50 |

Table 5: Pretraining Objectives in vision encoders.

| Encoder | Architecture | Policy VRAM (MB) | 150 step runtime (seconds) |
|---------|--------------|------------------|----------------------------|
| SAM | ViT-B | 568.317 | $7.076 \pm 0.724$ |
| DINOv2 | ViT-B | 336.244 | $4.995 \pm 0.701$ |
| CLIP | ViT-B | 573.281 | $2.307 \pm 0.070$ |
| MAE | ViT-B | 332.972 | $2.393 \pm 0.078$ |
| VIP | ResNet50 | 199.581 | $2.347 \pm 0.148$ |
| VC1 | ViT-B | 332.972 | $2.397 \pm 0.070$ |
| R3M | ResNet50 | 560.989 | $2.372 \pm 0.131$ |
| MVP | ViT-B | 332.972 | $2.409 \pm 0.063$ |
| MOCOv3 | ResNet50 | 262.767 | $2.117 \pm 0.126$ |
| IBOT | ViT-B | 657.525 | $2.401 \pm 0.069$ |
| OBJ RN | ResNet50 | 469.160 | $2.413 \pm 0.121$ |

Table 6: All policies can be trained on single A10 machines. Most policies take between 2 to 3 seconds to complete a trajectory and produce an action for 150 steps, however SAM and DINO take longer for their forward passes. Averages and standard deviations are reported over 5 repetitions on a single A10 machine.

## C   Reinforcement Learning Differences

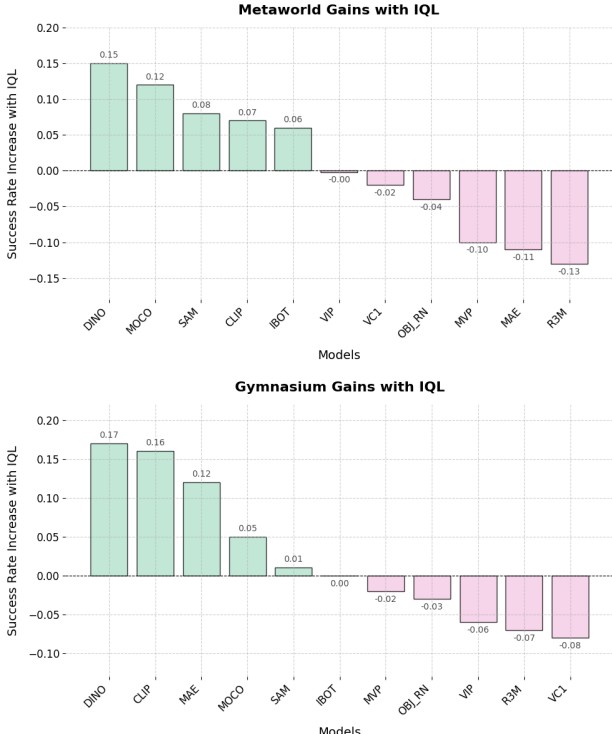

Figure 12: **Representations trained for manipulation perform better using behavior cloning.** In Gymnasium environments, MAE sees an improvement while in the Metaworld suite, MAE sees a decline in performance. Across the board, manipulation based representations see a decline in performance by using offline reinforcement learning.

## D    Sample Efficiency

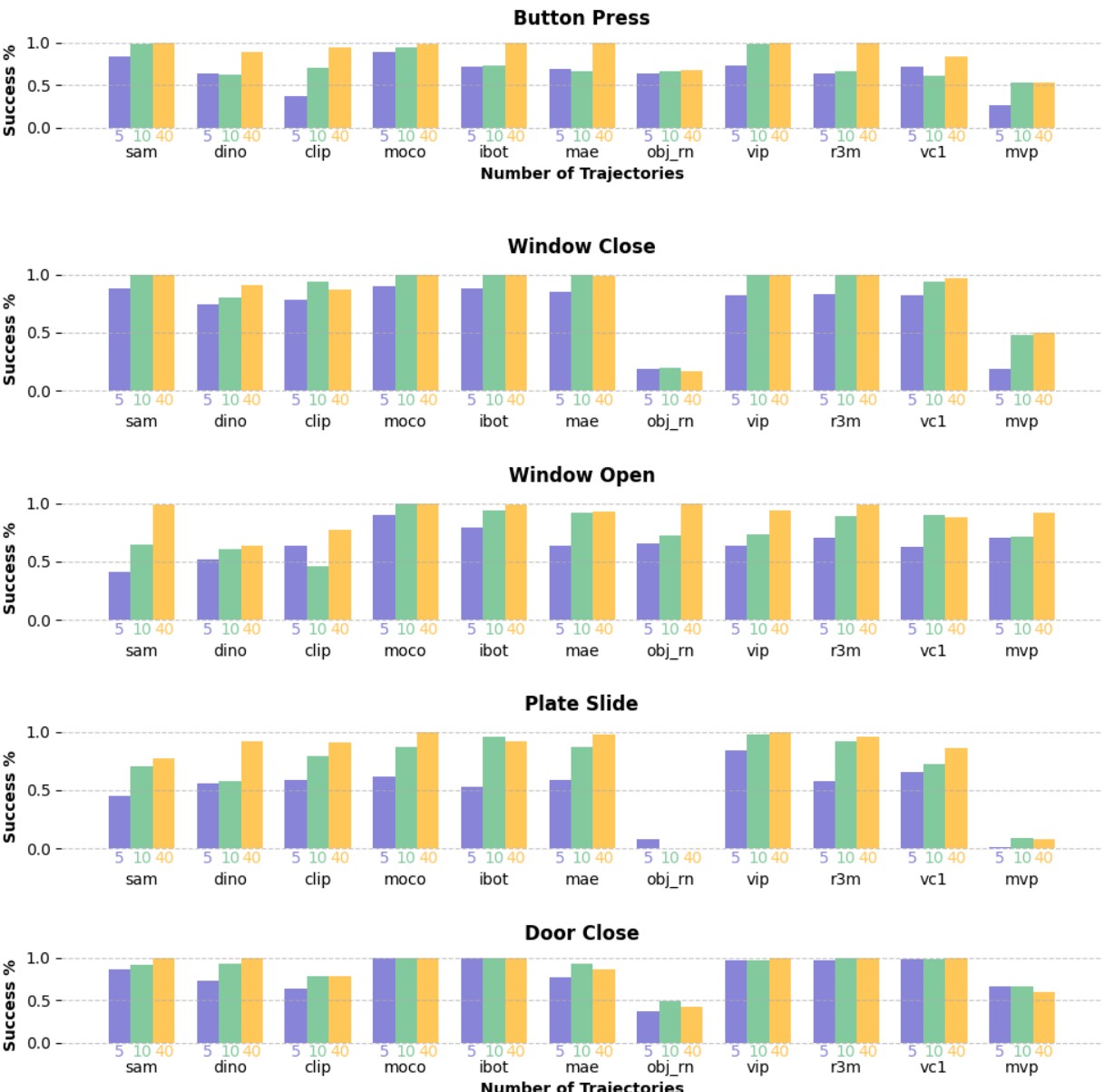

Figure 13: Sample efficiency of representations for behavior cloning evaluated on 5, 10, and 40 expert demonstrations. Representations trained for manipulation such as R3M, VIP are more sample efficient in general as they achieve higher success rates even with just 5 expert demonstrations.

# E   Key Position Prediction

For objects and goals, we observe similar correlation patterns. Figure 14 show a strong correlation between keypoint prediction and behavior cloning success rates.

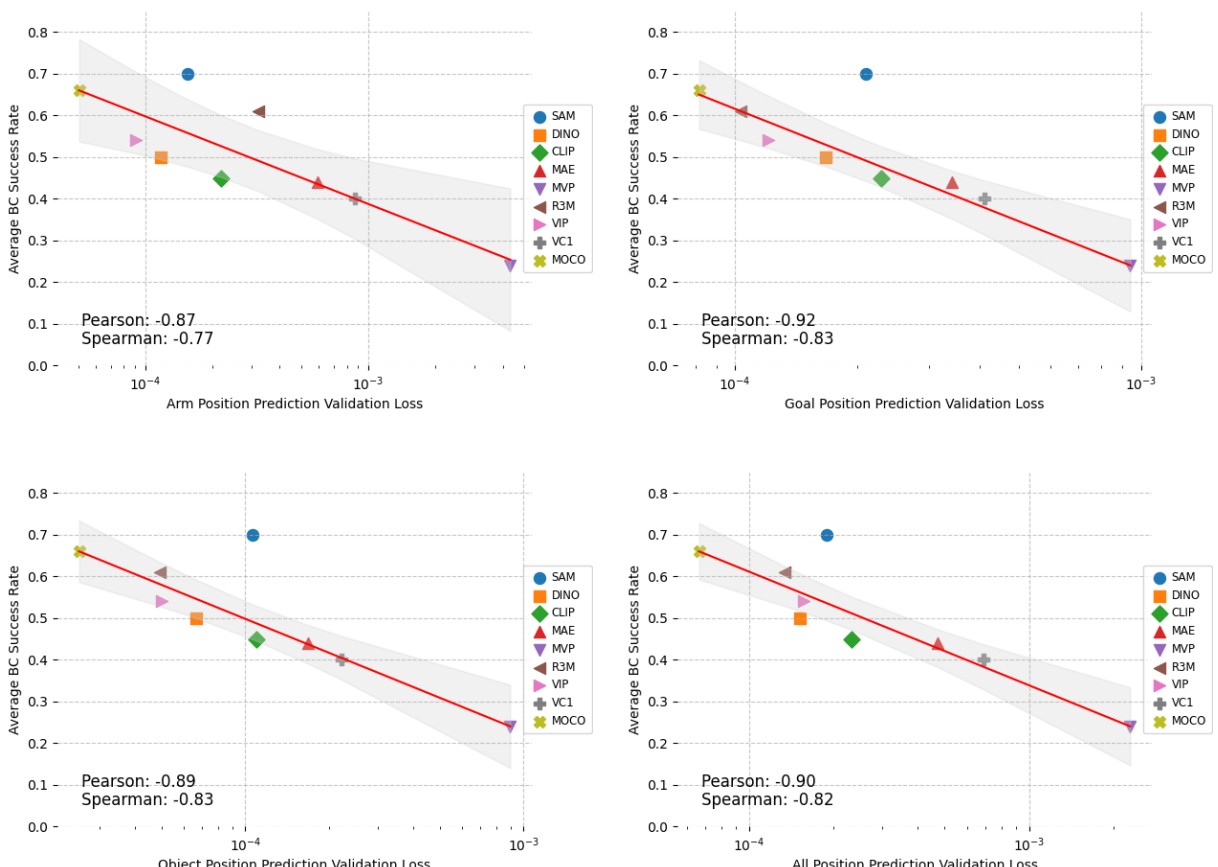

Figure 14: Correlations between predicting positions and behavior cloning success rates. All three demonstrate strong correlations.

| Position | Pearson (*p*-value) | Spearman (*p*-value) |
|---|---|---|
| Arm | -0.875 (0.0020) | -0.767 (0.0159) |
| Goal | -0.921 (0.0004) | -0.833 (0.0053) |
| Object | -0.886 (0.0015) | -0.833 (0.0053) |
| Combined | -0.904 (0.0008) | -0.817 (0.0072) |

Table 7: Pearson and Spearman rank Correlations for keypoint prediction are all statistically significant at the $\alpha = 0.05$ level

SAM can be considered as an influential observation and could affect the explanatory nature of our correlation analysis. The correlations without the SAM data points are in Figure 15. Additionally, a DFBETA analysis in Table 9 show that neither of the SAM observations have high influence (Kleinbaum et al., 1988).

The analysis in Tables 7 and 8 shows that removing SAM sees a maximum Pearson correlation increase of 0.05 and a maximum Spearman correlation increase of 0.17. While the Spearman increase of 0.17 is larger than the Pearson increase, Spearman correlation is a non-parameteric method, which inherently has a

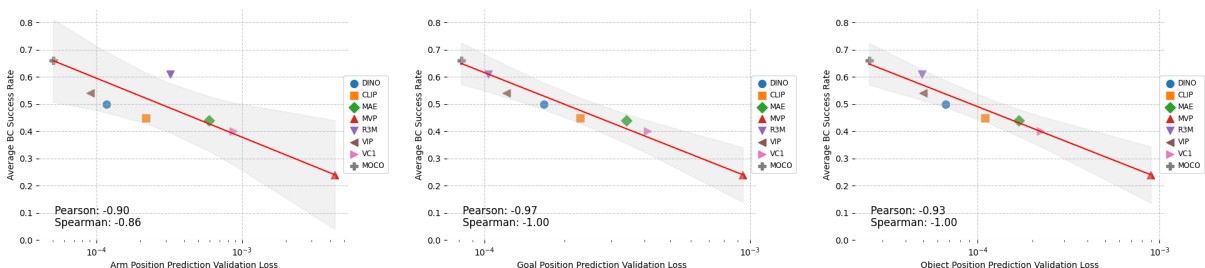

Figure 15: Correlations between predicting positions and behavior cloning success rates after removing SAM. Correlations remain strong.

lower statistical power compared to the Pearson correlation, thus its larger increase may not be statistically significant. The differences (Table 9) do not lie in the rejection region of $\Delta = 0.667$ and are not considered influential.

| Position | Pearson ($p$-value) | Spearman ($p$-value) |
|----------|---------------------|----------------------|
| Arm | -0.903 (0.0021) | -0.857 (0.0065) |
| Goal | -0.973 (0.0000) | -1.000 (0.0000) |
| Object | -0.933 (0.0007) | -1.000 (0.0000) |
| Combined | -0.937 (0.0006) | -0.976 (0.0000) |

Table 8: Without SAM, the Pearson and Spearman rank Correlations for keypoint prediction are also all statistically significant but are not too different from that in Table 7.

| | Arm | Goal | Object | Combined |
|---|-----|------|--------|----------|
| $DFBETA$ | 0.007 | 0.002 | 0.0003 | 0.007 |

Table 9: DFBETA analysis on the correlation coefficients by omitting SAM. All are below the threshold of 0.667, meaning SAM is not an influential point

## F  Clustering Representations

Interestingly, we find that grasping tasks are harder to solve than nudging tasks in both behavior cloning and reinforcement learning. We define grasping tasks to be tasks where the expert uses end-effectors of the robotic arm to pick up and grip an object in the environment, while a nudging task is one where the expert uses the end-effectors to push or drag an object without moving the effectors. Examples of grasping tasks are hammer, coffee pull, and bin picking. Nudging tasks include plate slide, soccer, and dial turn.

We aim to evaluate which representations benefit from using an offline reinforcement learning regime. To answer this question, we study the modality of the expert action distributions, following Ghasemipour et al. (2020); Ke et al. (2021). We view our expert data set as $(\phi(s), a)$ where the pretrained image representations $\phi(s)$ define the state space, and $a$ denotes the expert action. The state-action occupancy distribution is denoted by $\mu_E$. The expert policy is represented as $\pi^E(a|\phi(s))$ which is a conditional action distribution. We aim to learn the conditional action distribution using the expert collected trajectories with imitation algorithms. Prior work from Ghasemipour et al. (2020) differentiate between mode-covering and mode-seeking imitation algorithms that recover expert action distributions. Behavior cloning is a mode covering objective, which makes it particularly difficult to recover an expert action distribution that exhibits multiple modes.

We argue that the expert conditional action distributions $\pi^E(a|\phi(s))$ for grasping tasks contain more modes than for nudging tasks. This is clear from the optimization objective in behavior cloning with an $L_2$ objective, which is equivalent to minimizing a forward KL divergence of a Gaussian policy:

$$\min_\pi D_{KL}(\pi\|\pi^E) = \min_\pi \mathbb{E}_{a\sim\pi^E, s\sim\mu_E}\ \log\pi(a|\phi(s))$$

This formulation corresponds to a mode-covering objective. In contrast, the optimization procedure in implicit $Q$ learning follows an advantage-weighted regression objective, which Nair et al. (2020); Peng et al. (2019) demonstrate is equivalent to a reverse KL divergence:

$$\min_\pi D_{KL}\left(\pi^E\|\pi\right) = \min_\pi \mathbb{E}_{a\sim\pi^E, s\sim\mu_E}\ e^{\beta(Q(\phi(s),a)-V(\phi(s)))}\log\pi(a|\phi(s))$$

Unlike behavior cloning, this objective corresponds to a mode-seeking optimization regime, making it better suited for recovering optimal actions given a multi-modal expert action distribution.

Our expert conditional action distribution are samples drawn from a continuous state action occupancy $\mu^E$, and modes are detected through clustering (Chen et al., 2016). To test our hypothesis that grasping tasks exhibit higher modality in the conditional action distribution, we discretized the representation state space by first applying UMAP (McInnes et al., 2018) dimension reduction on the representations. UMAP preserves local density structure, makes non-linear down projections using intrinsic dimension and algebraic simplexes, and is widely used in dimension reduction and analyzing representations (Agarwal et al., 2021; Haarnoja et al., 2024; Dai et al., 2022). After applying UMAP, we use HDBSCAN (Campello et al., 2013) to cluster the state representations. Upon discretizing the representation space, we cluster actions within each discretized state representation, and count the number of clusters in the conditional action distribution and visualize an example in Figure 17. The details of this algorithm are given in Algorithm 1. We demonstrate that dimension reduction with UMAP is not necessary in Table 11.

For the results in Table 10, we use UMAP with 16 neighbors casting down to 3 dimensions. This produced clusters using HDBSCAN with the largest Silhouette values. We find that the number of negatively scored clusters is minimal with neighbors set at 16. We note that this method is sensitive to hyperparameter tuning.

When we do not reduce the dimensionality, and directly cluster with OPTICS to find approximate conditional action distribution, and then find clusters of actions, we still see that nudging tasks have less modes than grasping tasks in general across representations. The differences correlate with the gains in Figure 16 with Pearson $\hat\rho_r = 0.716$ and Spearman $\hat\rho_s = 0.7$. However, this method may not be preferred due to the computation time due to clustering in high dimensions.

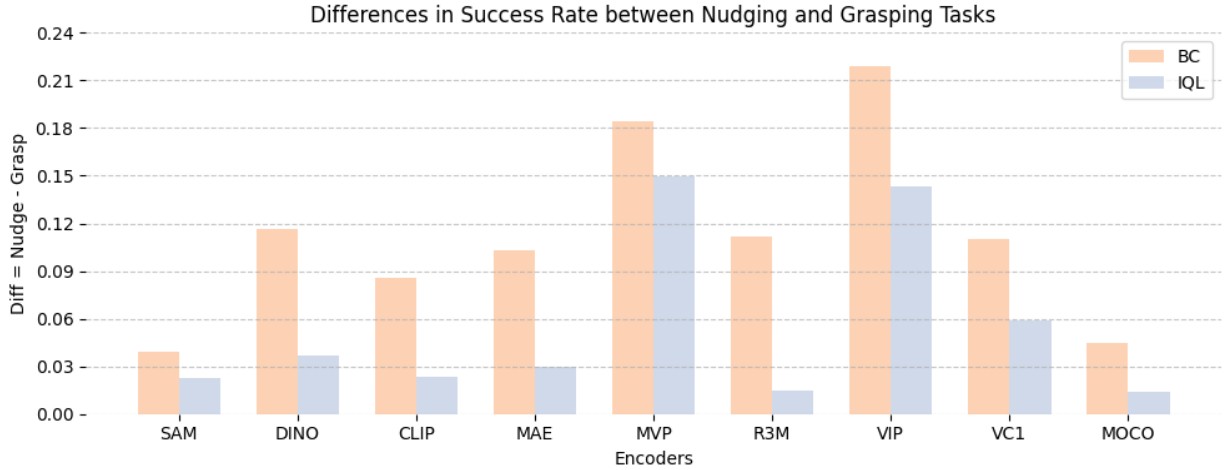

Figure 16: **IQL is better at solving grasping tasks than BC** The difference in success rates between grasping and nudging tasks vary across encoders, and also by the learning method. Reinforcement learning reduces the gap in success rate between nudging and grasping tasks (blue bar). However the level of improvement varies depending on representation.

---

**Algorithm 1** Determining modality with continuous expert state-actions

---

**Require:** Expert data $D^E = (\phi(s_t), a_t)_{1:N}$. Tuned clustering algorithm $K(\cdot) : \mathbb{R}^n \mapsto \mathbb{N}$, UMAP algorithm $U(\cdot) : \mathbb{R}^n \mapsto \mathbb{R}^d$

UMAP reduced embeddings $u \leftarrow U(\phi(s))$

Number of clusters $K \leftarrow K(u)$ and labels $K_L \in \{1, ..., K\}$

Append cluster labels to expert data $D^E \leftarrow (\phi(s_t), a_t, K_L)$

Initialize array $L$ to store number of modes in each cluster

**for** $c = 1...K$ **do**

    Subset $D^E$ to get states belonging to cluster $c$. $D_c \leftarrow (\phi(s_t), a_t, K_L = c)$

    Extract all actions $a_t$ from $D_c$ to get $A_c$

    Get number of clusters $M$ from conditional actions, $M \leftarrow K(A_c)$     ▷ Optional: Apply $A_c \leftarrow U(A_c)$

    Append $L[c] \leftarrow M$

**end for**

**return** Mean number of clusters $\bar{L}$

---

| Encoder $\phi(\cdot)$ | Average modality for Nudging | Average modality for Grasping | $\Delta$ |
|---|---|---|---|
| SAM | 3.405 | 4.083 | 0.678 |
| DINO | 2.987 | 3.929 | 0.942 |
| CLIP | 2.665 | 4.556 | 1.891 |
| MAE | 2.776 | 3.653 | 0.877 |
| MVP | 2.899 | 4.035 | 1.136 |
| R3M | 3.008 | 4.887 | 1.878 |
| VIP | 3.350 | 3.722 | 0.372 |
| VC1 | 2.709 | 4.317 | 1.608 |
| MOCO3 | 4.063 | 5.190 | 1.127 |

Table 10: **Average number of clusters in the conditional action distributions.** All conditional expert action distributions exhibit more modes in grasping tasks than for nudging tasks. Each number represents the average mode count from 3000 scenes representations in each of the 17 environments.

| Encoder $\phi(\cdot)$ | Modes in $\pi^E(a\|\phi(s))$ for nudging | Modes in $\pi^E(a\|\phi(s))$ for grasping | $\Delta$ |
|---|---|---|---|
| SAM | 3.81 | 3.88 | 0.07 |
| DINO | 2.75 | 5.89 | 3.14 |
| CLIP | 3.75 | 6.44 | 2.69 |
| MAE | 1.62 | 2.22 | 0.60 |
| MVP | 3.00 | 3.44 | 0.44 |
| R3M | 2.25 | 4.11 | 1.86 |
| VIP | 2.25 | 3.55 | 1.30 |
| VC1 | 2.63 | 3.44 | 0.81 |
| MOCO3 | 5.72 | 8.06 | 2.34 |

Table 11: Average number of clusters in the conditional action distributions. SAM representations form a state space that has almost the same number of clusters between nudging and grasping tasks. The clustering is done by the OPTICS (Ankerst et al., 1999) algorithm. This produces moderate to strong correlations

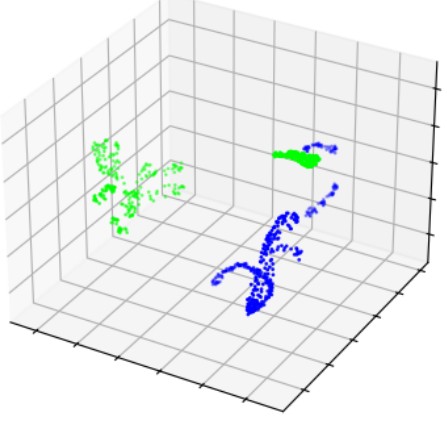

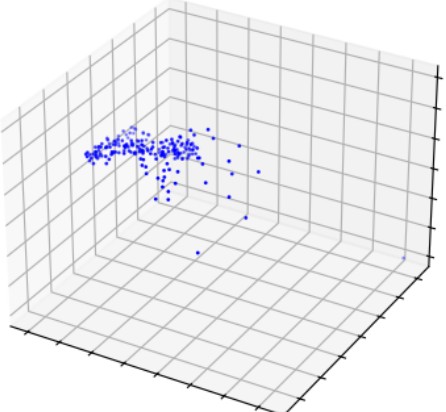

(a) 3D representation of expert actions from a given discretized $\phi(s)$ from coffee-pull (grasping), we can see multi-modal behavior in $\pi^E(a|\phi(s))$ in part due to the end effector movement.

(b) For a given discretized $\phi(s)$ from window-close (nudging), multi-modal behavior is not as prominent in $\pi^E(a|\phi(s))$. There is no end effector movement, and points roughly form a uniform subspace.

Figure 17: **3D representation of the action space.** The $(x, y, z)$ axes parameterize the arm movement, and the colors green and blue respectively parameterize the end effector being open and closed.

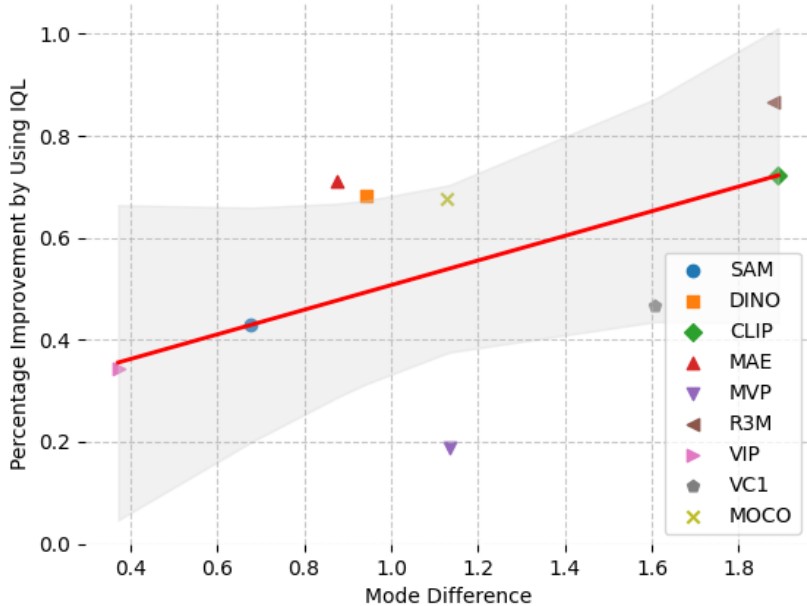

Figure 18: Correlations between the mode differences and the percentage improvements gained from using a mode seeking objective. The correlation is moderate and positive, which demonstrates the benefit of using mode seeking algorithms when the differences in expert conditional action distribution modes are larger.

## G Reconstructions

Reconstructions from various environments from Metaworld are shown in Figures 19 and 20 below:

### G.1 Reconstructions Correlate with Success Rates

We can also see the $p$-values in Table 12 and observe the correlations are significant at the $\alpha = 0.05$ level.

| Scene | Pearson ($p$-value) | Spearman ($p$-value) |
|---|---|---|
| Current | -0.805 (0.009) | -0.817 (0.007) |
| 5-step | -0.816 (0.007) | -0.817 (0.007) |
| 20-step | -0.809 (0.008) | -0.817 (0.007) |

Table 12: The correlation between image reconstruction is strongly correlated with behavior cloning success and the correlations are statistically significant at the $\alpha = 0.01$ level.

| Scene | Pearson ($p$-value) | Spearman ($p$-value) |
|---|---|---|
| Current | -0.910 (0.0001) | -0.918 (0.0000067) |
| 5-step | -0.932 (0.000029) | -0.873 (0.00045) |
| 20-step | -0.931 (0.000031) | -0.873 (0.00045) |

Table 13: The correlation between image reconstruction is even more strongly correlated with OOD behavior cloning success and the correlations are statistically significant at the $\alpha = 0.01$ level.

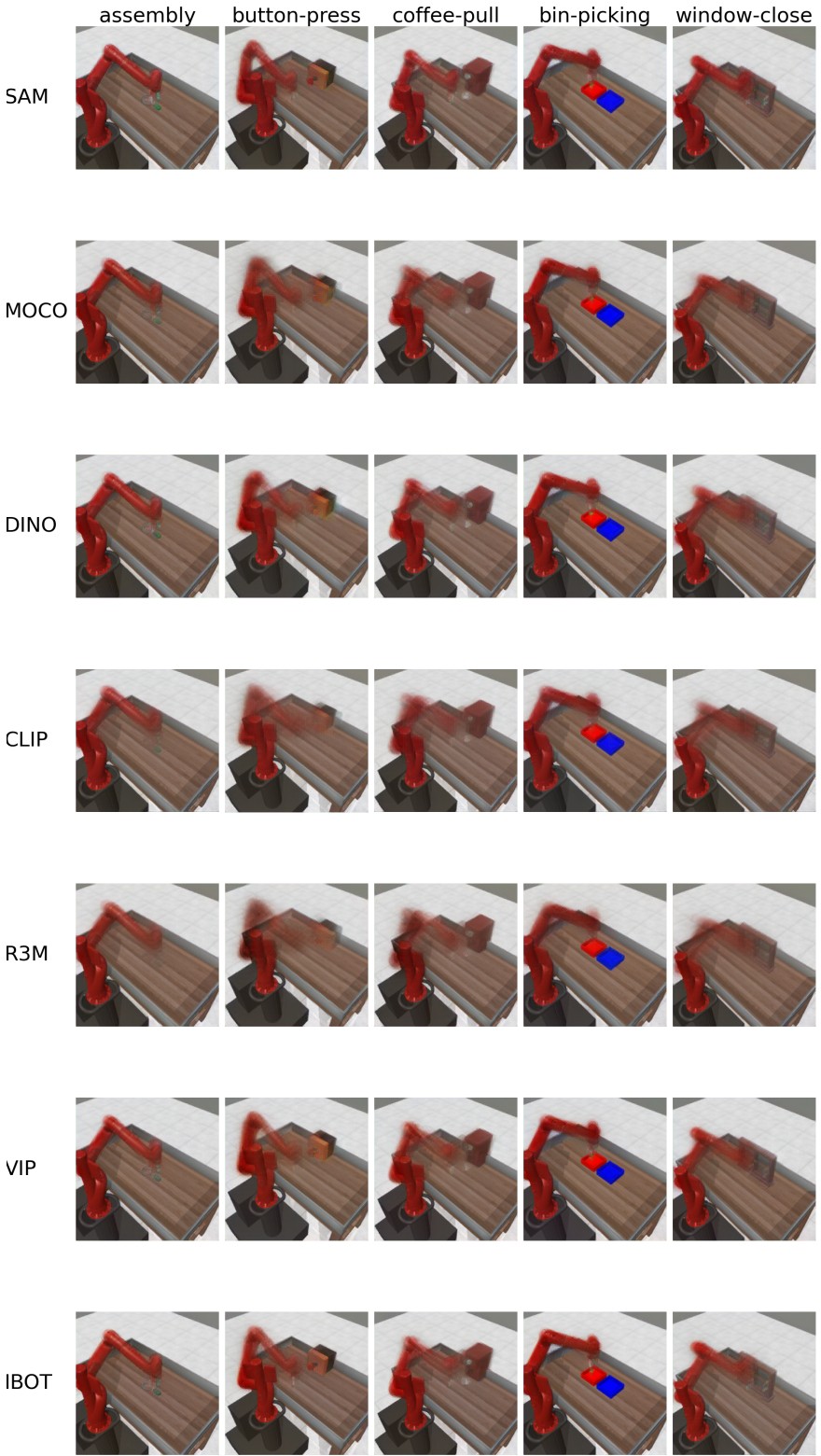

Figure 19: Non-reconstructive pretrained representations produced these reconstructions. Compared with Figure 20, there appears to be a higher level of fidelity when reconstructing the scene.

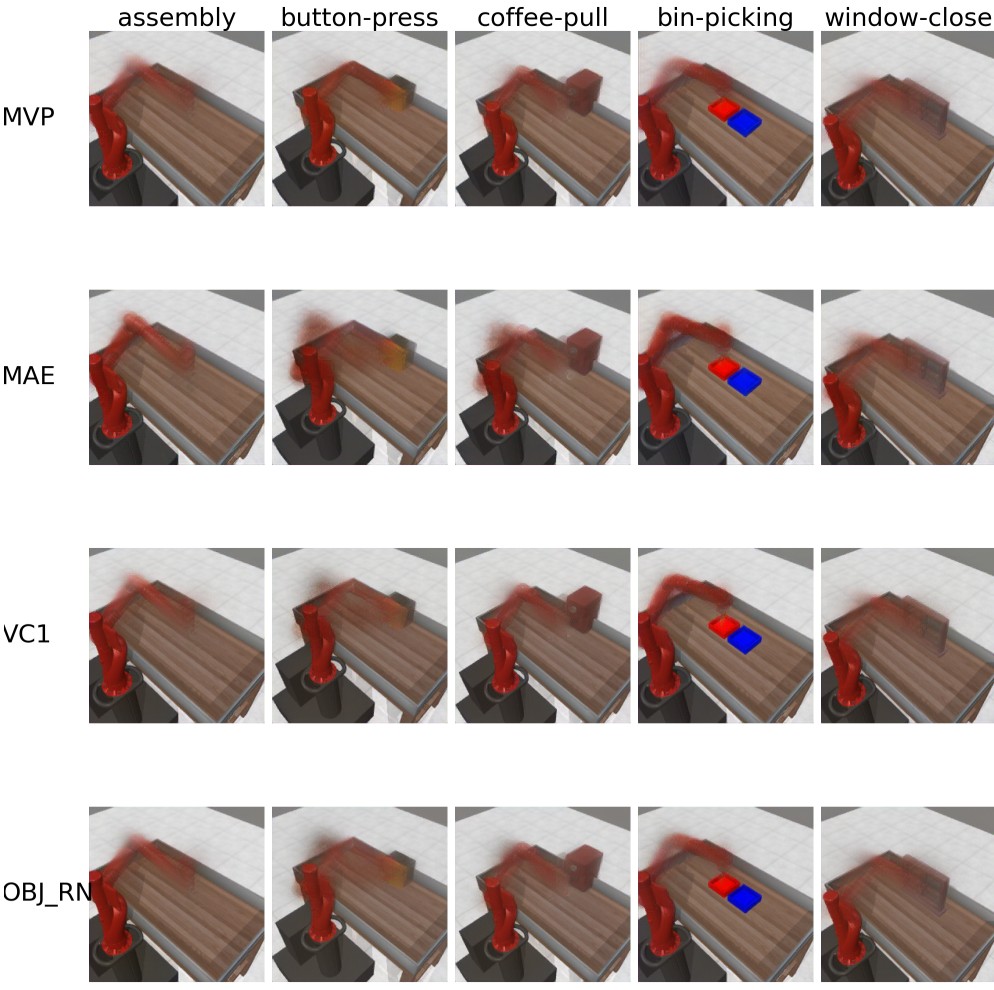

Figure 20: Reconstructive pretrained representations produced these reconstructions. Compared with Figure 19, there appears to be a lower level of fidelity when reconstructing the scene. In particular, the arm is not reconstructed very well, and objects in the scene are poorly reconstructed, if at all.

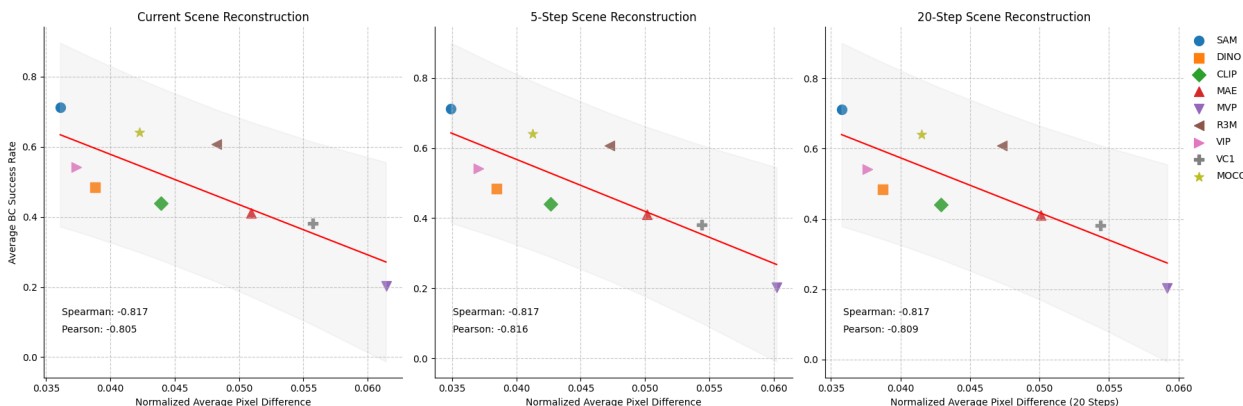

Figure 21: Pixel differences between reconstructions and the original scene rendering. Across the board, there is a strong negative correlation between reconstructions, future scene reconstructions, and the behavior cloning success rates.

# H   OOD Performance

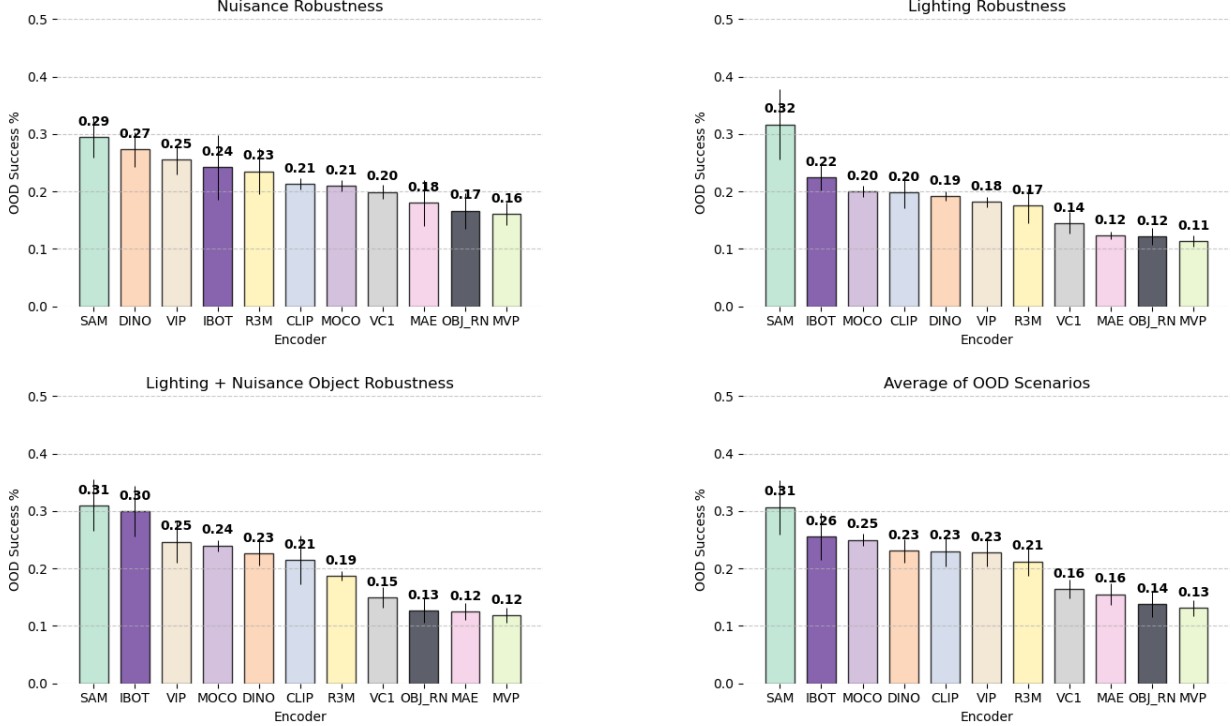

Figure 22: **Success rates accorss OOD performance is aligned with in distribution performance**. Across the board, all encoders perform worse in the OOD setting, but stronger encoders in the ID case still perform better in the OOD setting.

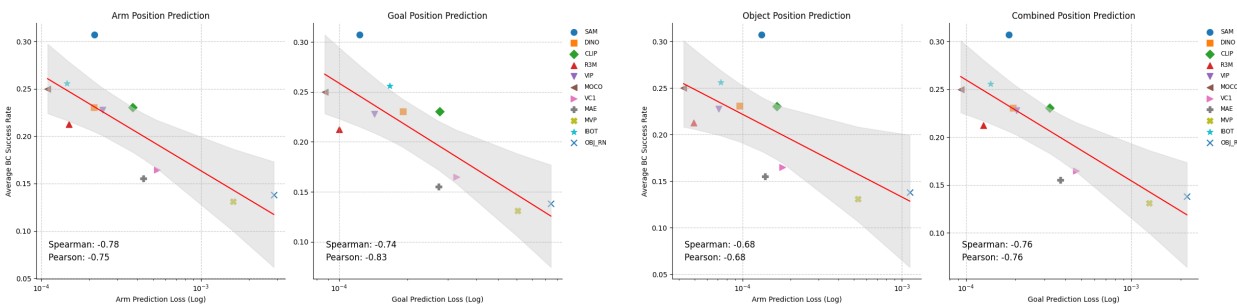

Figure 23: Keypoints correlate with OOD success rates

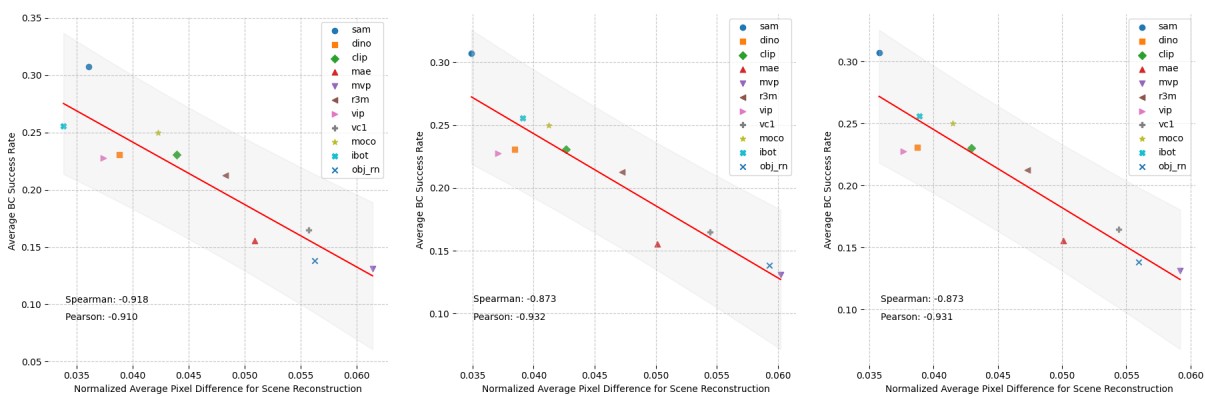

Figure 24: Reconstruction quality is strongly correlated with OOD success rates

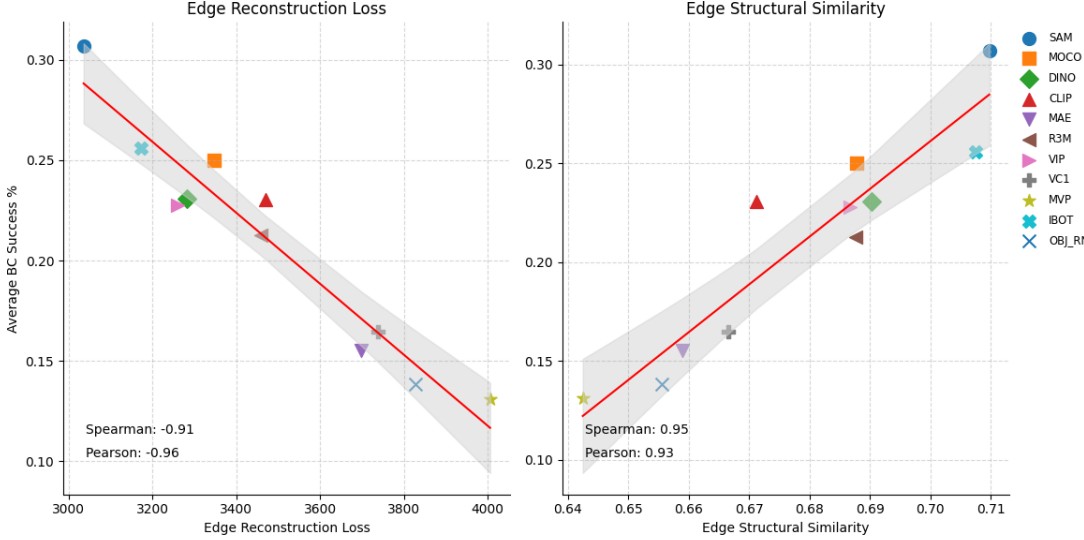

Figure 25: Edge reconstruction quality and structural similarity is strongly correlated with OOD success rates

## I   Attention versus MLP For SAM

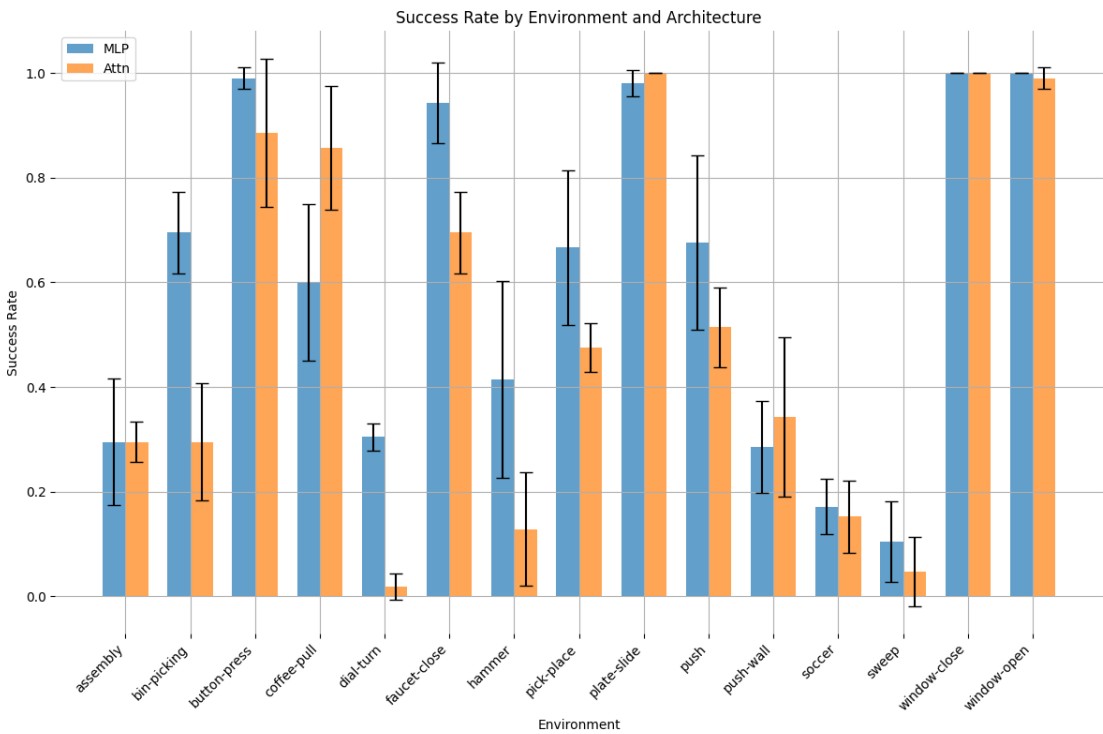

Figure 26: Attention based policy and MLP based policies perform very similarly for SAM

| Policy Architecture | Success % | Standard Deviation |
|:---:|:---:|:---:|
| MLP | 0.604 | 0.116 |
| Attention | 0.526 | 0.093 |

Table 14: The difference between MLP and Attention success rates are within one standard deviation of each other. This suggests the differences in success rate of MLP and Attention based policies are modest.

