# OpenReview forum: "Thoughts and Lessons on Using Visual Foundation Models for Manipulation"
_TMLR — Accepted by TMLR_

### Review · Reviewer_HXHd · 2025-03-02

**Summary Of Contributions:**

This work evaluates several pre-trained vision models as feature extractors for offline robotic learning, specifically, behavioral cloning and offline RL. The main takeaway from the evaluation is that vision models that maintain important features, such as edges, in their representations and can produce meaningful keypoints, are more effective for downstream manipulation tasks. In addition, the authors propose evaluation metrics that can assist in predicting a vision encoder performance for downstream manipulation tasks.

**Audience:**

Yes

**Claims And Evidence:**

Yes

**Requested Changes:**

* The conclusion that a good vision representation that maintains edges and has the ability to produce meaningful keypoints is very intuitive, and several previous works have used this kind of representation for decision-making tasks. Crucially, in many cases, this representation is self-supervised and object-centric. I highly recommend adding a discussion or referring to this line of works as they strengthen the main arguments in this paper.
See: https://arxiv.org/abs/2011.14381 , https://arxiv.org/abs/2404.01220v1  , https://arxiv.org/abs/2412.18907 , https://arxiv.org/abs/2410.23254 , https://arxiv.org/abs/2403.19578 , https://arxiv.org/abs/2106.07995
* Question: while SAM is not “reconstruction-based” per-se, it is usually referred to as object-centric, as it provides a (meaningful) decomposition of the scene into objects. Many object-centric vision models are trained with a reconstruction objective (see the above bullet). So, in that sense, would the “reconstruction-based representations are worse” conclusion still hold? I am not sure it is fair to conclude that “reconstruction objective is bad”, but perhaps a different kind of distinction between representations is required (one example is “holistic”/single-vector versus object-centric). Page 9, second paragraph, also emphasizes that the focus should rather be on object-centric representations (like SAM) that provide better reconstructions.
* Issues: my main two issues with this work are - (1) the choice of proper architecture to process the representations might make the conclusions invalid; (2) I think the conclusion about “reconstruction-based representations” might not necessarily be true, and it seems that the conclusion should be shaped around object-centric/non-object-centric according to the evaluated encoders.
* Question: if the conclusion is that “predicting keypoints can indicate the encoder’s performance”, why not use keypoints as the input representation? (this is actually done in previous works such as https://arxiv.org/abs/2106.07995 and https://arxiv.org/abs/2404.01220v1 but not evaluated here)
* Open-source code: do the authors plan to provide an open-source implementation of the benchmark and metrics for further research?
* See “Weaknesses” and “Minor” above.

**Strengths And Weaknesses:**

**Strengths**:
* Paper is clear and easy-to-follow.
* Many recent vision encoders are benchmark, indicating the great effort of the authors.
* In-depth analysis.
* Proposed metrics are intuitive and make a lot of sense for assessing a representation’s quality.
* Detailed appendix!

**Weaknesses**:
* Policy architecture: the choice of policy architecture might affect the downstream performance, and should be determined according to the representation structure (https://arxiv.org/abs/2203.05960 , https://arxiv.org/abs/2404.01220v1 ). For “holistic” representations (i.e., a single-vector representation for the entire image), an MLP might be a reasonable choice, but for more structured representations (e.g., tokens or object-centric representations), flattening the entire representation and using an MLP might not be the right choice (other architectures like deep sets, Graph Neural Networks or Transformers might be more reasonable, see above papers). “Don’t judge a fish by its ability to climb a tree” - flattening an entity/token-based representation and applying an MLP might not be a fair comparison to produce conclusions regarding the representation capabilities.
* Benchmark: simple environments and simple tasks. There are more challenging multi-object manipulation tasks (https://arxiv.org/abs/2410.20092 , https://arxiv.org/abs/2404.01220v1 ).
* Training data: it is worth noting that most of the pre-trained encoders are trained on real-world, natural, image data, where the pixel distribution is very different from the simulated pixels. As there are no real-world experiments, I would expect some discussion on the possible distribution shift (e.g., SAM might miss out-of-distribution objects).

**Minor**:
* I suggest removing the word “Thoughts” from the title of the paper as I find it a bit unappealing (but it is just my opinion).
* “Keypoints” and “Key points” appear interchangeably, I recommend using a single format (e.g. Abstract and page 2 paragraph 3).
* Broken Figure reference: Section 5.2, second paragraph, reference “Figure ??”, which I believe should be Figure 7.
* Figure 6: it is referenced nowhere in text and also requires better explanation, the caption is not clear w.r.t what is shown.
* Figure 7: it only shows the correlation with reconstruction and not with keypoint prediction. Did I miss something?
* Broken Figure reference: page 10, first paragraph: “Figure ?? and 9”.

---

> ### Author Response · Authors · 2025-04-10
>
> > [W1] Policy architecture: the choice of policy architecture might affect the downstream performance, and should be determined according to the representation structure
>
> We appreciate the reviewer for pointing out a crucial distinction here. While none of the evaluated representations are object-centric in the strict sense, that is, producing multiple distinct representations for multiple objects in the scene, SAM does stand out for producing patch-wise representations instead of one holistic vector for the whole scene. However, to ensure a fair apples-to-apples comparison across encoders, which almost all produce compact, scene-level holistic representations, we adopt a MLP policy architecture consistent across all experiments for all models. This MLP approach is widely used in prior works for assessing the quality of visual representations in downstream tasks [1,2,3], and is particularly effective for isolating the contributions of the representations [4] , and is widely used to benchmark visuomotor policies [5,6]. Despite this, we investigate whether attention-based policies could offer an advantage when used with SAM in Appendix I. Our results show that the attention-based policy is not conclusively better than MLP policies in our evaluation suite. Given that SAM already achieves the highest performance using the MLP policy, we opt to use the MLP policy across all encoders to preserve consistency in our evaluation. We agree that SAM fits in a special place since it does not produce compact embeddings, and we are prepared to remove SAM from the evaluation suite, if the reviewer determines this is necessary. If so, we have evaluated the IBOT encoder, which performs competitively as well. However, we believe that the SAM results are still interesting and important to show. If the reviewer determines that SAM should still be removed due to unequal footing, we are happy to place it into the appendix.
>
> [1] CLIP https://arxiv.org/abs/2103.00020
>
> [2] MVP https://arxiv.org/abs/2203.06173
>
> [3] MOCO https://arxiv.org/abs/1911.05722
>
> [4] Fine-Tuning can Distort Pretrained Features and Underperform Out-of-Distribution. ICLR 2022. https://arxiv.org/abs/2202.10054
>
> [5] The (Un)Surprising Effectiveness of Pre-Trained Vision Models for Control. ICML 2022. https://arxiv.org/abs/2203.03580
>
> [6] What Makes Pre-Trained Visual Representations Successful for Robust Manipulation?. CoRL 2024. https://arxiv.org/abs/2312.12444

---

> ### Author Response · Authors · 2025-04-10
>
> > [W2] Benchmark: simple environments and simple tasks. There are more challenging multi-object manipulation tasks
>
> We appreciate the reviewer’s point regarding the simplicity of our benchmark tasks relative to more complex, multi-object, or long-horizon manipulation settings. We understand that the evaluation environments could be seen as relatively simple. Our goal in selecting these tasks was to enable controlled comparisons across vision encoders while minimizing confounding factors such as planning complexity, object permanence, or long-horizon dependencies. This setup allows us to more directly study the relationship between visual representation quality and downstream control performance. We believe our diagnostic framework, particularly the link between edge/keypoint fidelity and control performance, will help guide such future investigations.

---

> ### Author Response · Authors · 2025-04-10
>
> > [W3] Training data: it is worth noting that most of the pre-trained encoders are trained on real-world, natural, image data, where the pixel distribution is very different from the simulated pixels. As there are no real-world experiments, I would expect some discussion on the possible distribution shift (e.g., SAM might miss out-of-distribution objects).
>
> We appreciate the reviewer’s observation regarding the distribution mismatch between the real-world data used to pretrain most vision encoders and the simulated environments used for our evaluation. Parisi et al 2021, demonstrate that pretraining data of vision encoders is not a core factor in manipulation success rates, and have further discussed this in Section 3.2. Furthermore, we evaluate encoder robustness under out-of-distribution (OOD) settings, which introduce visual domain shifts. Specifically, we perturbed the input distribution by adding nuisance objects in the scene, altering lighting conditions, and combining both nuisance objects and altering lighting conditions in the scene.
>
> Under these perturbations, we still find that the strong correlations between visual representation quality and manipulation performance persist—and in some cases, such as edge reconstruction and scene fidelity, the correlations are even stronger than in the original setting. This suggests that our findings are not solely tied to the specific pixel statistics of the training environments, and that key structural properties of representations, like edge and keypoint awareness, continue to play a significant role under distribution shift. We invite the reviewer to see the OOD analysis in Section 4.1, as well as correlations under the OOD setting in sections 5.1 and 5.2, and Appendix H.
>
> Diagnostic | Spearman | Pearson
> ------------|-------------|-----------
> Keypoint  | -0.750  | -0.790
> Reconstruction | -0.918 | -0.910
> Edge quality | -0.962 | -0.910

---

> ### Author Response · Authors · 2025-04-10
>
> > [MW]: Thank you for pointing out these typos and mislinks. They have been corrected in the updated draft.

---

> ### Author Response · Authors · 2025-04-10
>
> > [RC1]: The conclusion that a good vision representation that maintains edges and has the ability to produce meaningful keypoints is very intuitive, and several previous works have used this kind of representation for decision-making tasks. Crucially, in many cases, this representation is self-supervised and object-centric. I highly recommend adding a discussion or referring to this line of works as they strengthen the main arguments in this paper.
>
> We thank the reviewer for this valuable suggestion and we believe this broader framing also provides a good springboard for future works. In response, we have expanded our discussion to include recent object-centric and self-supervised works (e.g., EIT, EC-diffuser, Slot Transport, OCLR, etc). While our focus is on pretrained, frozen encoders that are not object-centric in the strict sense, we now highlight the connection between our proposed diagnostics and this parallel line of work. In particular, the importance of edge and keypoint fidelity in our analysis is consistent with the motivations behind object and entity centric learning and manipulation, highlighted in our related works section, as well as our discussion around keypoint analysis.

---

> ### Author Response · Authors · 2025-04-10
>
> > [RC2/RC3]: (1) the choice of proper architecture to process the representations might make the conclusions invalid; (2) I think the conclusion about “reconstruction-based representations” might not necessarily be true, and it seems that the conclusion should be shaped around object-centric/non-object-centric according to the evaluated encoders.
>
> Regarding (1):
>
>  We recognize that spatially structured outputs could potentially benefit from attention-based downstream policies. To explore this, we conduct an additional experiment using a transformer policy with SAM (Appendix I). This policy produces performance similar to an MLP based policy, and importantly, SAM already achieves the highest performance relative to other representations with the shared MLP policy. This MLP approach is widely used in prior work for assessing the quality of visual representations in downstream tasks, and is particularly effective for isolating the contributions of the representations, and is widely used to benchmark visuomotor policies. For consistency and comparability, we thus retain the MLP architecture across all models in the main evaluation.
>
> We would like to clarify that all but one of the encoders in our study produce compact, single-vector representations of scenes. This is a common design choice across many competitive and widely adopted pretrained vision models (e.g., R3M, VIP, MAE, DINOv2). To ensure a fair and controlled comparison, we adopt a shared MLP policy architecture for all models. This setup allows us to isolate the contributions of the visual representation itself, without introducing variability from differing downstream architectures.
>
> Although SAM is the only encoder in our suite that outputs a non compact representation, rather SAM provides dense patch-level embeddings and it does not learn to decompose scenes into discrete, object-level representations or latent slots. In that sense, SAM remains consistent with the scope of our analysis, which focuses on holistic representations rather than explicitly object-centric models.
>
> Regarding (2):
>
> We agree that one should take extra care in discussing what is meant by reconstructive objectives. In response, we have clarified in the paper that our conclusions around reconstructive objectives focus specifically on representations trained with holistic, scene-level reconstruction objectives (e.g., MAE, MVP), rather than structured, object aware reconstruction (e.g., MONet, Slot Attention). We have updated our terminology accordingly to avoid overgeneralization and to distinguish between global reconstruction and object-aware or slot-based representations.
>
> While SAM provides non-compact embeddings, they do not decompose scenes into object-level latents or use structured attention over discrete entities. Nonetheless, we agree that object-centric models are an important and growing research direction. In the revised version of the paper, we now cite several key works in this area and discuss their relevance to the broader topic of structured visual representations for control.
>
> Finally, we want to emphasize that even within our study of holistic, single-vector representations, there are clear and consistent trends: models trained with global reconstruction objectives underperform relative to those trained with discriminative or (self-)supervised objectives. We believe this insight remains valuable, particularly given how widespread single-vector representations remain in practice.

---

> ### Author Response · Authors · 2025-04-10
>
> > [RC4]: Question: if the conclusion is that “predicting keypoints can indicate the encoder’s performance”, why not use keypoints as the input representation?
>
> We thank the reviewer for raising this important question. Our goal in using keypoint prediction is not to propose keypoints as an input modality for downstream policies, but rather to use it as a diagnostic task for assessing the semantic and spatial precision of visual embeddings. Our contribution is to identify properties of pretrained, frozen visual encoders that make them effective for downstream manipulation, and we identify that keypoint prediction, among others, serves as a useful factor for analyzing this.

---

> ### Author Response · Authors · 2025-04-10
>
> > [RC5]: Open-source code: do the authors plan to provide an open-source implementation of the benchmark and metrics for further research?
>
> Yes, we intend to open source the code for further research.

---

> > ### Comment · Reviewer_HXHd · 2025-04-16
> > **Thank you for your clarifications and edits**
> >
> > I thank the authors or their effort, most of my concerns have been resolved. I still think that having "Thoughts" in the title is not a great choice. I have the other reviews and I tend to agree with their concerns as well, especially regarding the very limited and simulated benchmark. Finally, as I mentrioned in my original review, I'm not entirely convinced regarding the conclusions; however, I still think there is contribution in providing the "diagnostic tools" to assess the representation quality. As such, I'm leaning towards accept but will not argue against rejection if the other reviewers vote for it.

---

### Review · Reviewer_PDDW · 2025-03-12

**Summary Of Contributions:**

The submission investigates the effectiveness of various pre-trained visual foundation models specifically for robotic manipulation tasks, providing a evaluation across 21 diverse manipulation scenarios. The authors identify that a visual encoder's ability to reconstruct edges and accurately predict keypoints strongly correlates with downstream manipulation task success. The authors also introduce easy-to-compute diagnostic metrics (edge reconstruction and keypoint prediction accuracy) to predict the suitability of visual encoders for robotic tasks.

**Audience:**

Yes

**Broader Impact Concerns:**

None.

**Claims And Evidence:**

Yes

**Requested Changes:**

- Conduct OOD evaluation experiments to test the robustness and generalizability of the proposed diagnostic measures.

- Perform extensive ablation studies to isolate and clarify which aspects of edge and keypoint reconstruction contribute most to the enhanced manipulation performance.

**Strengths And Weaknesses:**

# Strengths:
- The paper perform extensive experiments across 21 diverse manipulation tasks, ensuring broad applicability and robustness of their findings.

- The paper identifies easily measurable diagnostic metrics (edge reconstruction quality and keypoint prediction accuracy) that strongly correlate with downstream task performance. This contribution has potential implications for efficiently selecting visual encoders in robotic applications.

# Weaknesses:
- The paper lacks detailed exploration into alternative auxiliary tasks or hybrid training approaches. Given the significance of the findings, an exploration of different pre-training combinations or regularizations would provide valuable insights.

- The paper primarily utilizes datasets with well-defined, structured environments. There is limited consideration of unstructured or more visually complex manipulation scenarios, which could challenge the identified correlations.

- The paper heavily relies on empirical observations without providing deep theoretical analysis or insights into why edge and keypoint reconstruction correlate with manipulation success.

---

> ### Author Response · Authors · 2025-04-10
>
> > [W1] The paper lacks detailed exploration into alternative auxiliary tasks or hybrid training approaches. Given the significance of the findings, an exploration of different pre-training combinations or regularizations would provide valuable insights.
>
> We agree this is an exciting direction. Our findings—particularly the strong correlation between edge/keypoint quality and task performance—can help inform future training regimes, but designing or evaluating new objectives is outside the scope of this analysis-focused work.

---

> ### Author Response · Authors · 2025-04-10
>
> > [W2]: The paper primarily utilizes datasets with well-defined, structured environments. There is limited consideration of unstructured or more visually complex manipulation scenarios, which could challenge the identified correlations.
>
> We appreciate the reviewer’s point. We understand that the evaluation environments could be seen as relatively simple. Our goal in selecting these tasks was to enable controlled comparisons across vision encoders while minimizing confounding factors such as planning complexity, object permanence, or long-horizon dependencies. This setup allows us to more directly study the relationship between visual representation quality and downstream control performance. We believe our diagnostic tasks regarding edge/keypoint fidelity and control performance will help guide such future investigations.

---

> ### Author Response · Authors · 2025-04-10
>
> > [W3] The paper heavily relies on empirical observations without providing deep theoretical analysis or insights into why edge reconstruction and keypoint reconstruction correlate with manipulation success.
>
> We appreciate the reviewer’s suggestion to investigate the technical insights into why edge reconstruction and keypoint analysis are correlated with manipulation success. Our work is grounded in empirical correlation analysis, and we do not make causal claims. Instead, we aim to provide diagnostic tools that can guide both practical representation selection and future theoretical modeling.

---

> ### Author Response · Authors · 2025-04-10
>
> >[RC2]: Perform extensive ablation studies to isolate and clarify which aspects of edge and keypoint reconstruction contribute most to the enhanced manipulation performance.
>
>
> We thank the reviewer for their comments and suggestions. Our work is designed as a systematic evaluation of pretrained vision encoders, with the goal of understanding what representational qualities matter most for downstream robotic manipulation. As such, we intentionally do not propose new architectures, auxiliary tasks, or training pipelines, but instead focus on controlled empirical analysis across a diverse encoder suite and 21 manipulation tasks.

---

> ### Author Response · Authors · 2025-04-10
>
> > [RC1] Conduct OOD evaluation experiments to test the robustness and generalizability of the proposed diagnostic measures.
>
>
> We thank the reviewer, and reviewer sUmQ for suggesting OOD analysis to enhance our work. In response, we have conducted OOD experiments by adding nuisance objects and lighting variations (Section 3.2 and Appendix H). In section 5, we also demonstrate that our diagnostic signals (edge/keypoint prediction) remain strongly correlated with success, even under OOD evaluation.

---

### Review · Reviewer_sUmQ · 2025-03-30

**Summary Of Contributions:**

This paper provides a thorough empirical study of nine pre-trained visual encoders across 21 diverse robotic manipulation tasks, splitting them into nudging versus grasping categories. Through both behavior cloning (BC) and offline reinforcement learning (RL), the authors uncover strong correlations between successful downstream performance and the ability of encoders to preserve keypoints and edges in image reconstructions. A key insight is that pre-training objectives that are discriminative (supervised or contrastive) generally yield better manipulator performance than purely reconstructive strategies. The work also highlights how multi-modal action distributions (especially in grasping tasks) can affect BC vs. offline RL outcomes. Overall, the paper offers valuable practical guidance on selecting and probing pre-trained encoders for manipulation tasks, shedding light on how task-relevant structural features in representations drive success.

**Audience:**

Yes

**Broader Impact Concerns:**

No obvious ethical or safety concerns arise.

**Claims And Evidence:**

Yes

**Requested Changes:**

Critical:
- Isolate backbone effects: Evaluate at least one reconstructive method on a ResNet and one discriminative method on a ViT to confirm whether performance differences are primarily tied to the objective or the architecture.
- Strengthen causal analysis: Conduct studies (e.g., artificially mask out edges or keypoints) or domain-randomize intensively to validate that these features are crucial for success and to mitigate reliance on correlated confounds.

Not critical but recommended:
- Real-world or domain-randomized validation: Evaluate how these encoders handle real-hardware noise, lighting changes, or simulated perturbations to support robust generalization claims.
- Detailed discussion of multi-modal behavior: Provide further analysis of how each encoder handles varying degrees of multi-modality in tasks (especially grasping), including more explicit comparisons of BC vs. offline RL outcomes.
- Practical deployment insights: Include memory usage, inference speed, or resource overhead for each encoder to guide researchers who wish to deploy these representations at scale.

**Strengths And Weaknesses:**

Strengths:
- Comprehensive benchmarking: The study covers 21 tasks, including nudging and grasping, using both BC and offline RL, offering broad evidence of the encoders’ generality and robustness.
- Actionable representation diagnostics: The authors propose easy-to-compute metrics (keypoint prediction, edge reconstruction) that strongly correlate with downstream policy success.
- Clarification on pre-training objectives: By reframing ‘supervised vs. self-supervised’ into ‘discriminative vs. reconstructive,’ the authors reconcile prior contradictory results in the literature.

Weaknesses:
- Mostly simulation-based: While the tasks are diverse, validating on real hardware or under stronger domain randomization would further confirm the encoder generalization claims.
- Lack of causal experiments: Although correlations are strong, more direct interventions (e.g., masking edges or keypoints) would help validate whether these features truly drive downstream success.
- No new learning algorithms introduced: The main innovation is empirical rather than algorithmic, which may limit the appeal for readers looking for novel RL or representation learning methods.
- Possible confusion between objective and architecture: Both reconstruction-based and discriminative methods use a mix of ResNet and ViT backbones, making it unclear if the observed performance differences are due to the pre-training objective or the network architecture.

---

> ### Author Response · Authors · 2025-04-10
>
> > [RC1/W4] Isolate backbone effects: Evaluate at least one reconstructive method on a ResNet and one discriminative method on a ViT to confirm whether performance differences are primarily tied to the objective or the architecture.
>
> We thank the reviewer for this helpful suggestion. In response to this comment, we have added an explicit evaluation to isolate the effects of architecture versus pretraining objective. Specifically, we compared:
> A reconstructive method on a ResNet backbone (OBJ RN)
> Reconstructive ViTs (MAE, MVP, VC1)
> A discriminative method on a ViT backbone (DINOv2, SAM, IBOT, CLIP)
> A discriminative ResNet (VIP, R3M, MOCOv3)
>
> As reported in Section 4.1 and 4.2, and summarized in Table 1 and Table 2, we observe no consistent performance differences attributable to architecture type. Both ViT and ResNet models perform at about the same level. In contrast, we observe a clear and consistent performance gap between discriminative and reconstructive pretraining objectives, regardless of backbone.
> These results strengthen our core conclusion about global reconstructive or discriminative pretraining objectives being a factor influencing downstream manipulation performance.
>
> Architecture | Objective & Success % | Standard Deviation
> --------------|---------|---------|----------
> ResNet | Holistic Reconstruction | 0.180 | 0.021
> ResNet | Discriminative | 0.603 | 0.049
> VIT	| Holistic Reconstruction | 0.360 | 0.086
> VIT	| Discriminative | 0.578 | 0.105

---

> ### Author Response · Authors · 2025-04-10
>
> > [W2/RC2] Lack of causal experiments: Although correlations are strong, more direct interventions (e.g., masking edges or keypoints) would help validate whether these features truly drive downstream success.
>
> We appreciate the reviewer’s suggestion to study causal mechanisms. However, we would like to clarify that the goal of this work is a correlational analysis rather than a causal one. We do not make any causal claims regarding whether ability to predict keypoints or reconstruct edges directly is a cause of better manipulation performance. Instead we aim to identify diagnostic signals in pretrained representations that consistently correlate with success across diverse environments and encoders.

---

> ### Author Response · Authors · 2025-04-10
>
> > [W1/RC3] Mostly simulation-based: While the tasks are diverse, validating on real hardware or under stronger domain randomization would further confirm the encoder generalization claims.
>
> We appreciate the reviewer’s observation regarding the distribution mismatch between the real-world data used to pretrain most vision encoders and the simulated environments used for our evaluation. Parisi et al 2021, demonstrate that pretraining data of vision encoders is not a core factor in manipulation success rates, and have further discussed this in Section 3.2. Furthermore, we evaluate encoder robustness under out-of-distribution (OOD) settings, which introduce visual domain shifts. Specifically, we perturbed the input distribution by adding nuisance objects in the scene, altering lighting conditions, and combining both nuisance objects and altering lighting conditions in the scene.
>
> Under these perturbations, we still find that the strong correlations between visual representation quality and manipulation performance persist—and in some cases, such as edge reconstruction and scene fidelity, the correlations are even stronger than in the original setting. This suggests that our findings are not solely tied to the specific pixel statistics of the training environments, and that key structural properties of representations, like edge and keypoint awareness, continue to play a significant role under distribution shift. We invite the reviewer to see the OOD analysis in Section 4.1, as well as correlations under the OOD setting in sections 5.1 and 5.2, and Appendix H.
>
> The below table summarizes the correlations between success rates and the proposed diagnostics.
>
> Diagnostic | Spearman | Pearson
> ------------|-------------|-----------
> Keypoint  | -0.750   	| -0.790
> Reconstruction | -0.918 | -0.910
> Edge quality | -0.962 | -0.910

---

> ### Author Response · Authors · 2025-04-10
>
> > [W3] No new learning algorithms introduced: The main innovation is empirical rather than algorithmic, which may limit the appeal for readers looking for novel RL or representation learning methods.
>
> We believe the findings are highly relevant to a broad set of researchers and practitioners such as robotics and RL practitioners who use pretrained encoders for visual representations, representation learning researchers who want to understand what makes visual features useful for control, or industry teams that are interested in determining the best vision encoders to use. Our goal is to contribute insights that can inform the design of future algorithms, rather than necessarily introducing a new method. We hope the clarity and consistency of our empirical findings provide value to the community.

---

> ### Author Response · Authors · 2025-04-10
>
> > [RC4] Detailed discussion of multi-modal behavior: Provide further analysis of how each encoder handles varying degrees of multi-modality in tasks (especially grasping), including more explicit comparisons of BC vs. offline RL outcomes.
>
> This analysis was carried out in Appendix F, where we demonstrate how each encoder performs in both nudging and grasping tasks. Table 16 shows the comparison between BC and Offline RL outcomes, stratified by encoder.

---

> ### Author Response · Authors · 2025-04-10
>
> > [RC5] Practical deployment insights: Include memory usage, inference speed, or resource overhead for each encoder to guide researchers who wish to deploy these representations at scale.
>
> Thank you for this suggestion. We will incorporate these statistics in the final draft of the paper.

---

### Decision · Action_Editor_cBfS · 2025-05-12

**Recommendation:** Accept as is

**Comment:**

All reviewers thought positively of the the revised version of the paper, and recommended acceptance.

**Audience:**

Yes

**Claims And Evidence:**

Yes, all three reviewers are in agreement that the claims are supported by accurate, convincing, and clear evidence.